# Polarised cell intercalation during *Drosophila* axis extension is robust to an orthogonal pull by the invaginating mesoderm

**Claire M. Lye** *, **Guy B. Blanchard, Jenny Evans, Alexander Nestor-Bergmann, Bénédicte Sanson** *

Department of Physiology, Development and Neuroscience, University of Cambridge, Cambridge, United Kingdom

* cmg38@cam.ac.uk (CML); bs251@cam.ac.uk (BS)

## Abstract

As tissues grow and change shape during animal development, they physically pull and push on each other, and these mechanical interactions can be important for morphogenesis. During *Drosophila* gastrulation, mesoderm invagination temporally overlaps with the convergence and extension of the ectodermal germband; the latter is caused primarily by Myosin II–driven polarised cell intercalation. Here, we investigate the impact of mesoderm invagination on ectoderm extension, examining possible mechanical and mechanotransductive effects on Myosin II recruitment and polarised cell intercalation. We find that the germband ectoderm is deformed by the mesoderm pulling in the orthogonal direction to germband extension (GBE), showing mechanical coupling between these tissues. However, we do not find a significant change in Myosin II planar polarisation in response to mesoderm invagination, nor in the rate of junction shrinkage leading to neighbour exchange events. We conclude that the main cellular mechanism of axis extension, polarised cell intercalation, is robust to the mesoderm invagination pull. We find, however, that mesoderm invagination slows down the rate of anterior-posterior cell elongation that contributes to axis extension, counteracting the tension from the endoderm invagination, which pulls along the direction of GBE.

## Introduction

The generation of tissue shapes during animal development is complex, but we are beginning to understand how cell autonomous behaviours such as oriented cell division, cell shape changes, and cell intercalation contribute to tissue morphogenesis. In addition to intrinsic forces generated within the cell, it is becoming increasingly clear that developmental morphogenesis can also be influenced by extrinsic forces acting at the scale of the tissue, generated by the deforming tissue itself or by the movements of neighbouring tissues [1,2]. For example, extrinsic forces in developing tissues can cause cell shape changes, drive or reorient cell intercalations, reorganise planar polarity, or lead to changes in gene transcription [3–12]. In addition to examples where physical interactions between tissues are important for their

---

**Data Availability Statement:** Relevant data are within the paper and its Supporting information files, and code can be found at https://zenodo.org/records/10887088.

**Funding:** This work was supported by two Wellcome Trust Investigator Awards to BS (099234/Z/12/Z and 207553/Z/17/Z). ANB was supported by a University of Cambridge Herchel Smith Fund Postdoctoral Fellowship. The funders had no role in study design, data collection and analysis, decision to publish, or preparation of the manuscript.

**Competing interests:** The authors have declared that no competing interests exist.

**Abbreviations:** AP, antero-posterior; DV, dorso-ventral; GBE, germband extension; HCR, hybridisation chain reaction; *ind, intermediate neuroblast defective*; LRR, Leucine-rich repeat; MRLC, Myosin II regulatory light chain; *sim, single-minded*; *vnd, ventral nervous system defective*.

morphogenesis, it is becoming apparent that mechanisms also exist to buffer physical forces and mechanically isolate tissues from one another [13–15].

Early embryogenesis in *Drosophila* has become an important paradigm for understanding how tissue morphogenesis is driven by a combination of forces generated directly by cells (intrinsic forces) and, indirectly, at the tissue or embryo scale (extrinsic forces), as several well-characterised morphogenetic movements occur within a short period of time [16–18]. Specifically, at the same time as the mesoderm invaginates ventrally and the endoderm invaginates posteriorly, the ectoderm initiates convergence and extension to extend the main body axis of the embryo (germband extension (GBE)) (Fig 1A–1E). GBE is caused primarily by polarised cell intercalation driven by Myosin II [19–22]. Antero-posterior (AP) patterning controls the planar polarised distribution of junctional cortical Myosin II, which leads to shrinkage of cell junctions parallel to the dorso-ventral (DV) axis (so called DV-oriented junctions), causing cells to exchange neighbours [23–26]. In addition to this primary mechanism, an extrinsic AP-oriented pull from the invaginating posterior endoderm contributes to axis extension by driving elongation of cells in AP [3,4]. We showed by quantitative analysis that the elongation of cells along AP contributes to about one-third of tissue extension during the first 30 minutes of GBE (the so-called "fast" phase of extension [23]), while polarised cell intercalation contributes to the other two-third [3]. In addition to elongating cells, the AP-oriented extrinsic force produced by endoderm invagination orients growing junctions during polarised cell intercalation [9].

While the role of endoderm invagination in GBE is well established, it is unclear what the impact of mesoderm invagination is, if any, as the results reported so far are inconsistent [3,27–30]. The autonomous cell behaviour underlying mesoderm invagination (and endoderm invagination) is apical constriction, which requires the assembly of a contractile network of actomyosin cytoskeleton at the apex of the cells [31,32]. During mesoderm invagination, the ventral-most mesodermal cells contract the strongest, making a ventral furrow through which the whole of the mesoderm tissue invaginates (Fig 1A, 1C and 1E). Because of the geometry of the mesoderm (an AP elongated rectangle, Fig 1A), these ventral cells contract predominantly in DV causing a DV-oriented pull, which stretches the more lateral mesodermal cells along DV [33–38]. It is unclear how much this DV-oriented tension propagates to the adjacent ectoderm. It has been proposed that stretching of the lateral mesoderm (and also dorsal tissue) accommodates the mechanical tension induced by the invaginating mesoderm, providing a mechanical buffer for the ectoderm [13,35]. However, contrary to the idea that the germband is buffered from a mechanical impact from mesoderm invagination, other studies suggest that mesoderm invagination speeds up GBE [3,28–30]. In particular, it has been proposed that the rate of polarised cell intercalation increases in response to mesoderm invagination leading to enhanced planar polarisation of Myosin II, through a mechanotransduction mechanism [28,30]. If the ectoderm is indeed pulled by the invaginating mesoderm, another possible outcome, which has not been proposed so far, is that mesoderm invagination could act a brake for GBE, since the mesoderm pulls against the direction of convergence and perpendicularly to the direction of extension.

The aim of this study is to evaluate systematically the different possible impacts of mesoderm invagination on GBE. Some of the discrepancies reported can be explained by the challenge of quantitatively comparing wild-type embryos with mutant embryos defective for mesoderm invagination. Here, we tackle this difficulty by acquiring imaging datasets that are as comparable as possible between wild-type and *twist* mutant embryos. We use these datasets to quantify different metrics, including Myosin II densities and junctional behaviours (Fig 1F). We first ask whether the mesoderm mechanically interacts with the extending ectoderm. We next investigate whether a DV-oriented pull by the mesoderm could augment Myosin II

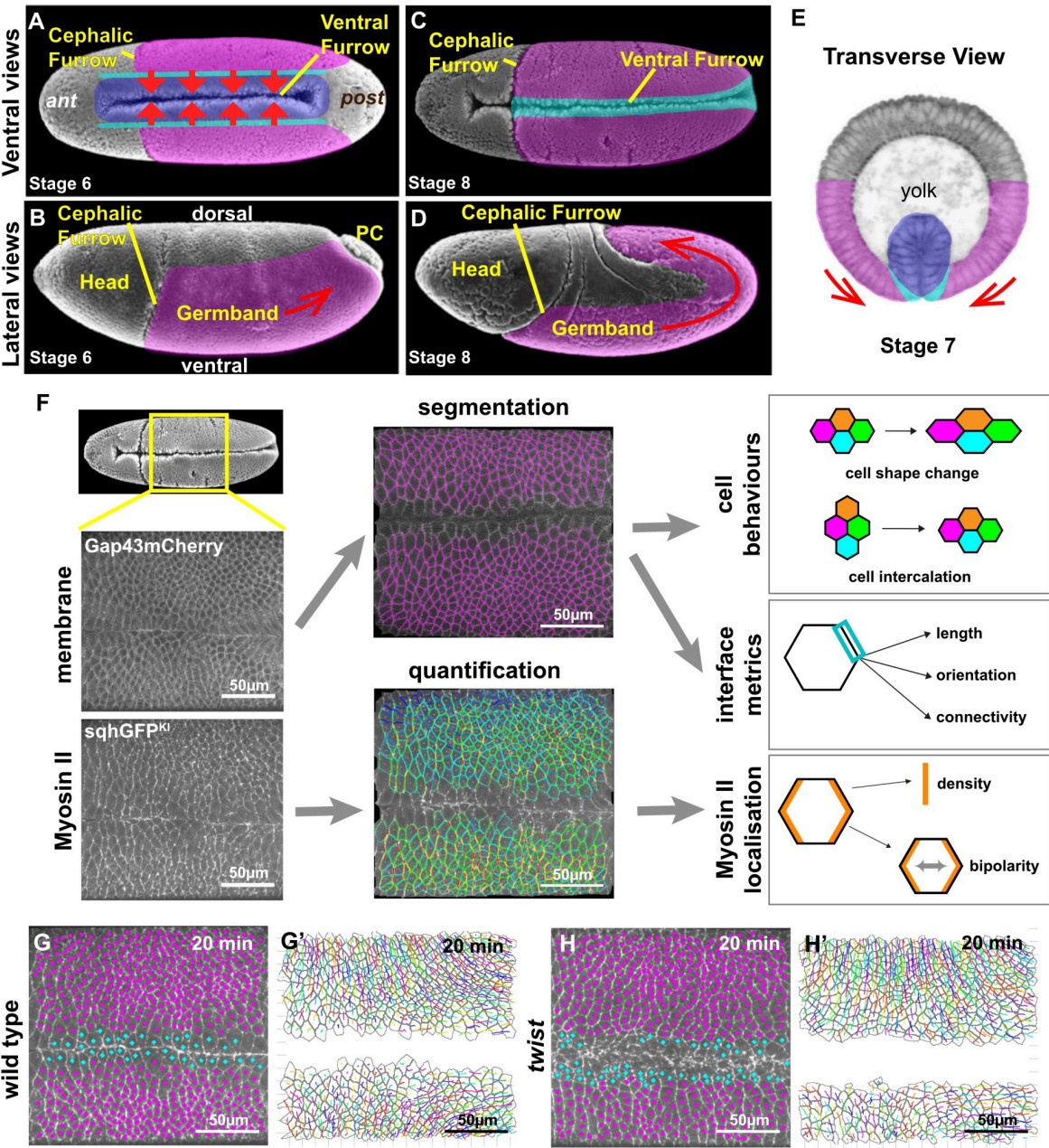

**Fig 1. Investigating the impact of mesoderm invagination on GBE.** Scanning electron micrographs from Flybase (**A**-**D**) and tranverse section micrograph (from Munoz [55]) (**E**) showing anatomy of *Drosophila* embryos from early to mid-GBE (stages 6 to 8). Coloured overlays highlight the converging and extending ectodermal germband (magenta), mesectoderm (cyan), and mesoderm (blue). Red arrows depict tissue movements. The mesoderm invaginate ventrally, while the ectodermal germband starts convergence and extension. The endoderm invaginates where the pole cells are located, at the posterior end of the germband. Anterior to the left in (**A**-**D**). (**F**) Workflow diagram summarising image acquisition and analysis. Panels are maximum intensity projections of live spinning-disc confocal imaging of Gap43Cherry (upper panel) and sqhGFP$^{KI}$ (lower panel), followed by segmentation and tracking of Gap43Cherry channel to extract cell behaviours and interface metrics and quantification of Myosin II channel to extract Myosin II density on cell interfaces (blue, low; red, high) and bipolarity measures. (**G** and **H**) Example frames for defining cell types in wild-type (**G**) and *twist* (**H**) movies (mesoderm/mesectoderm in cyan, ectodermal germband in magenta) overlayed on Myosin II channel (maximum intensity projection). Unmarked cells are those poorly tracked and excluded from the analysis. See also S1 Fig and S3 and S4 Movies. (**G**' and **H**') Examples of segmentation of ectodermal cells (grey) and tracking of cell centroids (coloured lines) in wild type (**G**') and *twist* (**H**'), showing trajectories of cells over the previous 5 minutes. ant, anterior; GBE, germband extension; PC, pole cells; post, posterior.

enrichment at DV-oriented cell junctions via a mechanotransduction mechanism, and if this could increase the rate of junctional shrinkage during neighbour exchange events. Alternatively, if no such mechanotransduction occurred, then as DV-oriented cell junctions stretch in response to mesoderm invagination, cortical Myosin II might become diluted. Additionally, changing the balance of extrinsic forces could have mechanical effects on the speed of junctional shrinkage and growth throughout the process of cell neighbour exchange. The change in boundary conditions could also change how much cells elongate along AP in response to endoderm invagination. Finally, mesoderm invagination might help align cells, and subsequent neighbour exchanges, with the embryonic axes, so that neighbour exchanges drive maximal tissue convergence and extension.

## Results

### Comparing germband extension in wild-type and *twist* mutant embryos

To investigate whether mesoderm invagination has an impact on GBE, we carried out a quantitative analysis of GBE in wild-type and *twist* mutants, which lack mesoderm invagination. We chose to analyse *twist* mutants rather than *snail* or *twist snail* mutants, because although ventral furrow formation fails in all these mutants, some contractility remains in mesodermal cells in *twist* mutants alone, which decreases the width of the mesoderm and makes the ventrolateral field of cells more comparable with wild type. In *snail* or *twist snail* mutants, no contractility remains and the uninvaginated mesoderm takes significant space at the surface of the embryo [31].

We acquired movies of the ventral side of embryos from before the onset of mesoderm invagination until the middle of GBE, for both wild-type and *twist* mutants (Fig 1F–1H'). Four wild-type and 6 *twist* movies were analysed throughout this study. We labelled cell membranes with Gap43mCherry, and Myosin II with GFP-tagged knock-in of MRLC to visualise all Myosin II molecules (called *sqhGFP*$^{KI}$ here; [39]). Strains were constructed to ensure that these labels were expressed at the same level in wild-type and *twist* embryos (Methods). We also controlled the temperature during imaging (20.5 +/− 1°C) to limit variability. Cell contours were segmented from the Gap43mCherry channel and tracked over time (Methods and S1 and S2 Movies) to calculate metrics describing key cell and interface behaviours (Fig 1F) [40]. Myosin II density and planar polarity were quantified from the Myosin II channel as before (Fig 1F) [26].

Our field of view captured the convergence and extension of the ectodermal tissue on the ventrolateral surface of the embryo but also included mesodermal cells and mesectodermal cells. The inclusion of the ventral midline enabled us to precisely measure orientations with respect to this landmark (Methods). To restrict our analysis to the extending ectoderm, we carefully excluded mesodermal and mesectodermal cells based on (i) whether they had invaginated (wild type only); (ii) their proximity to the midline; and (iii) the timing of their cell divisions [41,42] (Fig 1G and 1H' and S1, S3 and S4 Movies).

Next, we checked that the ectodermal cells analysed in wild type and *twist* are comparable in terms of DV and AP genetic patterning. We examined the expression patterns of key DV patterning genes: *single-minded* (*sim*) is expressed in the mesectoderm, ventral nervous system defective (*vnd*) is expressed in the ectoderm abutting *sim*, and intermediate neuroblast defective (*ind*) is expressed next, abutting *vnd* [43]. While there is some de-repression of both *sim* and *ind* in the non-invaginated mesoderm and in the mesectoderm in *twist* mutants, the expression of all 3 genes is similar in the ectoderm (S2 Fig). We also checked the expression patterns of the Leucine-rich repeat (LRR) cell surface receptors Tolls 2, 6, 8, and Tartan, which are required for the polarised distribution of Myosin II during GBE downstream of the AP

patterning genes [22,44–47]. We confirmed that these genes are expressed with the same patterns in *twist* mutants as in wild type (S2 Fig). Therefore, the regions of the ectodermal germband we analyse here in wild type and *twist* mutants are indistinguishable in term of patterning.

To compare averaged metrics between wild type and *twist* mutants, we synchronised the movies in time, using the start of tissue extension along the AP axis (measured as AP tissue strain rate) in each movie as time zero as before (Methods) (S3A and S3B Fig) [3,4,26,47]. To check that timelines of wild-type and *twist* mutant movies were comparable once synchronised, we checked the timings of 2 developmental events: (i) when Myosin II becomes detectable apically in the ectoderm; and (ii) when the first cell divisions occur (Methods). We found that appearance of apical ectodermal Myosin II (around 10 minutes before GBE) and of the first cell divisions in the lateral ectoderm (around 32 minutes after GBE) are grouped in time across movies (S3C Fig). We note that mesectoderm cell divisions are delayed by nearly 10 minutes in *twist* mutants compared to wild type, which might be a consequence of misregulation of mesodermal and mesectodermal patterning genes. We conclude that our movie synchronisation gives comparable developmental time windows for the 2 genotypes.

We next determined the number of cells tracked at each time-point in wild-type and *twist* movies (S3D–S3F Fig). At early time points, we track a total of approximately 800 cells (from 4 movies) for wild type and a total of 2,000 cells (from 6 movies) for *twist*. By the end of the period of analysis, over 2,500 cells are included for both wild type and *twist*.

In summary, the above characterisation demonstrates that we have acquired comparable live imaging datasets of thousands of tracked cells of the ectodermal germband for wild-type and *twist* embryos covering a developmental window of 15 minutes prior to 30 minutes after the onset of GBE. We confirmed that DV and AP patterning in the extending ectoderm is unchanged in *twist* mutant compared to wild type, allowing us to explore the mechanical impact of mesoderm invagination on GBE independently of patterning.

## The invaginating mesoderm pulls on the extending ectoderm

As the mesoderm invaginates, cells of the ectodermal germband undergo translation ventrally in wild type, indicating that these 2 epithelial tissues are mechanically connected. This movement towards the midline is much reduced in *twist* embryos as expected from the lack of invagination (Fig 2A and 2A'). This mechanical linkage suggests that the invaginating mesoderm pulls on the adjacent ectodermal germband, generating a DV-oriented tensile stress. Gradients of cell elongation and apical area increase have been found as a signature of epithelial tissues being subject to tension from extrinsic forces generated by neighbouring morphogenetic movements [3,4,30,35]. To ask whether the ectoderm is being pulled by the invaginating mesoderm, we analysed these metrics for all ectodermal cells in the field of view for 4 wild-type and 6 *twist* embryos (Methods and Fig 2).

In wild-type embryos, the rate of change in apical cell lengths along DV (DV cell shape strain rate, Fig 2B and 2C) and in apical areas (Fig 2E) both increase significantly from 10 minutes before the start of GBE, showing that cells are starting to stretch. The rate of DV stretching increases rapidly, peaking at about 3 minutes into GBE, before rapidly decreasing until 7 to 9 minutes into GBE (Fig 2C and 2E). After this time, the cells start shrinking, reducing their DV length and cell area until the end of the period analysed, 30 minutes into GBE. To compare the timing of these cell behaviours in the ectoderm to the behaviour of the mesoderm, we documented the key events of mesoderm invagination in our 4 wild-type movies (S4A Fig). We find that the period of cell stretching corresponds to when the mesoderm starts contracting and then invaginates in the movies (about -13 to +7 minutes, yellow bar in Fig 2C and 2E).

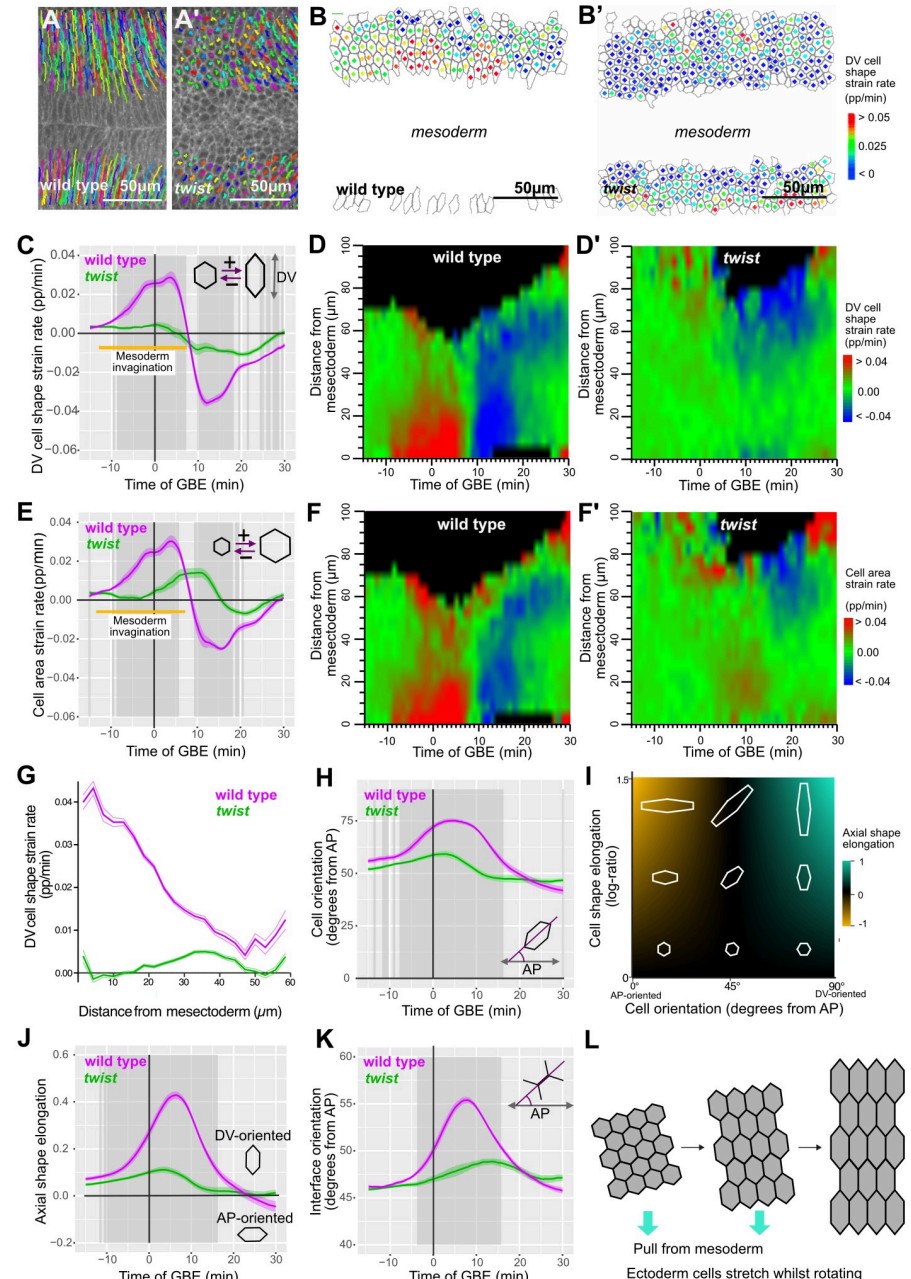

**Fig 2. Ectodermal cells during GBE are stretched and rotate in response to ventral pull from mesoderm invagination.** (**A**, **A'**) Cell trajectories (coloured lines) of germband cells over 5–10 minutes of GBE overlaid on Gap43Cherry maximum intensity projection at 10 minutes of GBE. Anterior to the left, ventral centrally. (**A**) wild type: germband cells move significantly ventrally and begin to move posteriorly. (**A'**) *twist*: germband cells move only slightly ventrally and begin to move posteriorly. (**B**, **B'**) Analysis of cell shape strain rate, projected along DV axis (proportion/minute) in ectodermal cells in wild type (**B**) and *twist* (**B'**) at 2.5 minutes into GBE. Cell shape strain rate of each cell is indicated by coloured square at the centre of each cell (cell outlines, grey). (**C**) DV-projected cell shape strain rate in proportion/minute (abbreviated as pp/min throughout the manuscript) of ectodermal cells, summarised for wild type (magenta) and *twist* (green) against time of GBE. The average time-period of mesoderm invagination, from first sustained apical constriction until all mesoderm cells are internalised, is indicated with a yellow bar here and in (**E**). Timing of mesoderm invagination was assessed by eye for all wild-type movies (S4A Fig). (**D** and **D'**) DV projected cell shape strain rate (pp/min) of germband cells, summarised for wild type (**D**) and *twist* (**D'**) against time of GBE and distance from mesectoderm (μm) (see Methods). (**E**) Cell area strain rate (pp/min) of germband cells, summarised for wild type (magenta) and *twist* (green) against time of GBE. (**F** and **F'**) Cell area strain rate (pp/min) of ectodermal cells, summarised for wild type (**F**) and *twist* (**F'**) against time of GBE and distance from mesectoderm

(μm) (see Methods). (**G**) Average DV cell shape strain rate from −7.5 to 7.5 minutes of GBE summarised for wild type (magenta) and *twist* (green) against distance from mesectoderm (μm). Thinner lines show standard error. (**H**) Average orientation of principal axis (degrees from AP axis) of ectodermal cells against time of GBE for wild type (magenta) and *twist* (green). Note that if cells are randomly oriented, then the average cell angle would be 45 degrees. (**I**) Our "axial shape elongation" measure is a combined measure of cell elongation and cell orientation, giving a value of 1 for cells strongly elongated in DV, minus 1 for cells strongly elongated in AP and 0 for isotropic cells or elongated cells at 45 degrees to embryonic axes (see Methods). (**J**) Axial shape elongation measure for ectodermal cells against time of GBE for wild type (magenta) and *twist* (green). (**K**) Average orientation of cell interfaces from the AP axis of ectodermal cells for wild type (magenta) and *twist* (green) plotted against time of GBE. (**L**) Summary of average ectodermal cell behaviour in response to tensile force from invaginating mesoderm (note that angle of rotation shown is arbitrary). Throughout the manuscript: Dark grey shading indicates periods of significant difference ($p < 0.01$; see Methods), and ribbons show standard error between genotypes; in explanatory diagrams of the measures being plotted on graphs (for example, Fig 2C), the + directional arrow indicates where values will be positive, and the − direction arrow indicates where values will be negative. Data associated with this figure can be found in S1 Data. AP, antero-posterior; DV, dorso-ventral; GBE, germband extension.

This is consistent with previous reports of cell stretching in the ectoderm in response to meso-derm invagination [3,4,27,30]. We next plotted cell shape changes and apical area changes on spatiotemporal plots to examine how they evolve in time and space. In wild type, the rate of change for DV cell shape and apical area have the same spatiotemporal pattern (red signal in Fig 2D and 2F), confirming that the cells are stretching in DV and indicating that the cells are under DV-oriented apical tension. Also, there is a gradient in the observed cell stretching from ventral to lateral, suggesting that the source of tension is positioned ventrally, as expected if the mesoderm is pulling (Fig 2G). Consistent with this notion, in *twist* embryos, cell stretching in DV is virtually abolished (Fig 2C–2F') and there is no gradient of DV cell elongation (Fig 2G). We conclude that the invaginating mesoderm generates a tensile force that is transmitted across the germband, causing ectodermal cells to stretch along DV, with a ventral to dorsal gradient. In wild type, following this period of intense cell stretching in DV, cells shrink again along this axis (Fig 2C), eventually regaining an isotropic shape [3].

This tissue-level tensile force could also cause ectodermal cells to reorient, so we quantified cell orientation (Fig 2H–2K). First, we plotted cell orientation over time and find for both wild type and *twist* that cells are slightly DV-oriented already 15 minutes before the start of GBE (average angle greater than 45 degrees, Fig 2H). This trend increases rapidly in wild type until about 5 minutes into GBE, reaching a maximum average angle of 75 degrees, showing that cells become very DV-oriented. Cells in *twist* mutants also become briefly more aligned with the DV axis, but to a much lesser extent than wild type, peaking at about 60 degrees. A com-bined measure of cell elongation and cell orientation, which we call "axial cell elongation bias," confirms that in wild type, cells rotate to become more DV-oriented as they stretch in DV (Fig 2I and 2J and Methods). The strongest effect is seen from 5 minutes prior to the start of GBE to 15 minutes after (peaking around 7 minutes) and is mostly absent in *twist*. This demon-strates that mesoderm invagination causes cells not only to stretch but also to realign perpen-dicular to the ventral midline. We also plotted average orientation of cell junctions from the AP axis (Fig 2K). At −15 minutes before GBE starts, the average orientation is approximately 45 degrees for both wild type and *twist*, showing that there are no bias in directionality of inter-faces just before mesoderm invagination. In wild type, the average orientation increases towards DV from −10 minutes as cells stretch in DV, peaking at 8 minutes at approximately 55 degrees, consistent with DV-oriented interfaces becoming more aligned with the DV axis as cells stretch. In contrast for *twist*, the average orientation changes little over time (Fig 2K), consistent with the finding that there is little DV stretch in cells in *twist* mutants.

Extrinsic tissue-level forces can drive cells to move past each other, leading to passive cell intercalations [5,6]. To ask whether the DV-oriented pull of mesoderm invagination causes

cell slippage in the ectoderm, we quantified the cell intercalation strain rate as previously [40] (Methods). We find that the intercalation strain rates along the DV axis is negligible or negative during the period of DV cell stretch, with no difference between wild type and *twist* mutants (S4B Fig). This demonstrates that the cell stretching caused by mesoderm invagination in the ectoderm is not accompanied by passive cell rearrangement.

In summary, our results show that in wild type, the contraction force from the invaginating mesoderm causes the ectodermal cells of the germband to stretch in DV and change orientation, without causing cells to rearrange along the axis of tension (Fig 2L). We next asked whether this mechanical force exerted by the invaginating mesoderm onto the ectoderm impacts on axis extension.

### Comparing Myosin II planar polarisation in wild-type and *twist* mutants

Myosin II–driven polarised cell intercalation is the main cell behaviour causing the convergence and extension of the *Drosophila* germband [18,22]. During this process of cell neighbour exchange, DV-oriented junctions between AP cell neighbours shrink, so that these cells lose contact with each other, and then a new cell junction grows in the perpendicular orientation (Fig 3A). Cell junction shrinkage is driven by Myosin II, which is enriched in shrinking junctions under the control of the AP patterning genes [19,20]. In addition to genetic regulation, there is some evidence in the *Drosophila* germband that Myosin II can be recruited through a mechanosensitive feedback mechanism [24,48]. Since mesoderm invagination generates a DV-oriented tensile force within the ectoderm, mechanosensitive recruitment or stabilisation

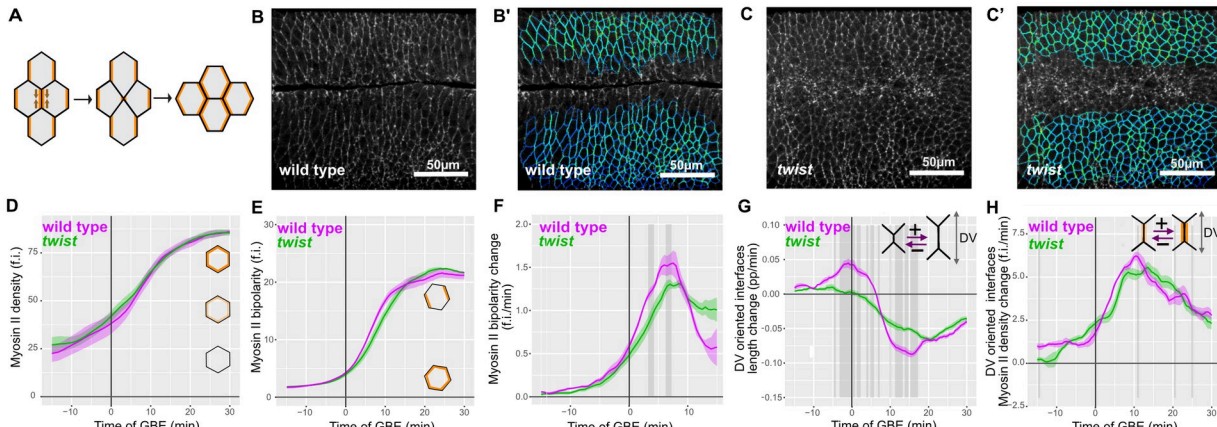

**Fig 3. Myosin II bipolar enrichment in wild-type and *twist* mutants.** (**A**) Diagram showing Myosin II–driven cell intercalation. Myosin II (orange) shows planar polarised localisation, being enriched in a bipolar manner on the DV-oriented junctions (between cells neighbouring each other along the AP axis). Interface shrinkage (brown arrows) leads to a transient 4-way vertex, followed by growth of a new AP-oriented interface. (**B** and **C**) Normalised Myosin II signal at the level of adherens junctions extracted from confocal image stacks for quantification in wild type (**B**) and *twist* (**C**) at 10 minutes of GBE. See Methods and S7 Fig for details of extraction and normalisation of signal. (**B'** and **C'**) Quantified interface Myosin II signal of ectodermal cells corresponding to images in (**B**) and (**C**) for wild type (**B'**) and *twist* (**C'**). Dark blue corresponds to lowest Myosin II density through to red for highest Myosin II density. Note there are no interfaces at this time point with highest Myosin II density (red), so green-yellow shows highest Myosin II in these frames. (**D**) Average density of Myosin II on ectodermal cell interfaces, summarised for wild type (magenta) and *twist* (green) against time of GBE. (**E**) Unprojected bipolarity of Myosin II of ectodermal cell interfaces, summarised for wild type (magenta) and *twist* (green) against time of GBE (f.i. denotes fluorescent intensities throughout the manuscript). (**F**) Rate of change of unprojected bipolarity of Myosin II of ectodermal cell interfaces, summarised for wild type (magenta) and *twist* (green) against time of GBE. (**G**) Proportional rate of interface length change, for DV-oriented (oriented more than 45 degrees from the AP axis) interfaces only, summarised for wild type (magenta) and *twist* (green) against time of GBE. (**H**) Rate of Myosin II density change, for DV-oriented (oriented more than 45 degrees from the AP axis) interfaces only, summarised for wild type (magenta) and *twist* (green) against time of GBE. Data associated with this figure can be found in S2 Data. AP, antero-posterior; DV, dorso-ventral; GBE, germband extension.

of Myosin II might contribute to the enrichment of Myosin II on DV-oriented cell junctions. Alternatively, cortical Myosin II might be diluted along DV-oriented cell junctions as they are stretched by the invaginating mesoderm. We therefore asked how Myosin II concentration on DV-oriented junctions is affected by mesoderm invagination.

First, we quantified fluorescence as a readout of Myosin II density at the level of the adherens junctions in the ectoderm, for our wild-type and *twist* movies (Fig 3B and 3C') (Methods). We plotted average fluorescence intensities for these cell–cell interfaces over time and observed that in both wild type and *twist*, Myosin II density starts low before GBE (Fig 3D), consistent with our previous measures [26]. Myosin II levels then increase, gradually at first and then more rapidly until approximately 15 minutes after the start of GBE, after which they continue to increase more slowly. Wild-type and *twist* curves are virtually identical demonstrating that accumulation of Myosin II at apical cell–cell junctions is unaffected in *twist* mutants (see also S3C Fig).

Myosin II is expected to be enriched on membranes oriented approximately parallel to the DV embryonic axis [19,20], and this is apparent in our movies of both wild type and *twist* (see S3 and S4 Movies). We quantified this Myosin II planar polarisation by measuring Myosin II bipolar distribution around the membrane of each cell as before [26]. Because there are differences in cell orientation between wild type and *twist* (see Fig 2H–2K), we plotted the average bipolarity of germband cells, independently of the orientation of the bipolarity. In both wild-type and *twist* embryos, this unprojected Myosin II bipolarity measure starts to increase around the start of GBE, increases quickly in the first 15 minutes of GBE and then stays high for the remaining 15 minutes, with no statistically significant difference between the 2 genotypes (Fig 3E). Spatiotemporal plots for these data show that these temporal patterns are similar for all cells in the field of view and that there is no obvious ventral to dorsal gradient in wild type (S4D and S4D' Fig). Although there is no significant difference in unprojected bipolarities at any time between the 2 genotypes, the *rate* of increase appeared marginally faster between 2 and 12 minutes of GBE in the wild type compared to *twist* mutants (Fig 3E). Therefore, we calculated the rate of change of Myosin II bipolarity, and we found that in *twist* embryos, it is slightly reduced compared to wild type, showing some statistically significant differences around 5 minutes of GBE (Fig 3F).

Because these differences are seen while cells are stretching in DV (see Fig 2C–2F'), we asked whether cell interface stretching is associated with an increased rate of Myosin II recruitment. We plotted the proportional rate of stretching of DV-oriented interfaces and confirm that significant stretching occurs in wild type, but not in *twist*, between approximately −10 and 7 minutes of GBE (Fig 3G). We then plotted the rate of Myosin II density change for DV-oriented interfaces and observed that the rate of Myosin II recruitment increases rapidly during this time-period not only in wild type but also in *twist* mutants whose DV-oriented interfaces undergo very little stretching (Fig 3H). Apart from some very brief bursts of significance, there is no difference between wild type and *twist* for this measure (Fig 3H). Altogether, this indicates that if there is any increase in Myosin II localisation early in GBE in response to the physical pull by the mesoderm on the ectoderm, it is at the limit of what we can detect with our measures. Interestingly, our results show, however, that Myosin II density is maintained in DV-oriented junctions that are stretching under tension from mesoderm invagination, compared to unstretched DV-oriented junctions in *twist* embryos, indicating that a homeostatic mechanism maintains Myosin II cortical density during junctional stretching.

## Comparing the rates of junctional shortening and neighbour exchange in wild-type and *twist* mutants

In addition to intrinsic forces within the germband driving its convergence and extension, ectoderm extension is facilitated by an extrinsic pull (along AP) from the posterior endoderm [3,4,9]. Both the convergence and the extension in the germband could be mechanically counteracted by the pull along DV from the invaginating mesoderm. If so, in the absence of mesoderm invagination, we might expect junctional shortening to speed up due to the lack of a DV tension opposing junctional shrinkage. Thus, boundary conditions imposed by the endoderm and mesoderm invaginations could affect the behaviour of ectodermal cells during GBE. Junctional shortening rates could also be influenced by mesoderm invagination in the opposite way (i.e., junctional shortening being faster in wild type than in *twist* mutants) through mechanosensitive Myosin II recruitment, as considered above.

Therefore, we examined whether mesoderm invagination impacts on the rate of junctional shrinkage of the extending ectoderm throughout early and mid-GBE (0 to 30 minutes). We find that while there is a consistent trend over time for the proportional rate of junctional shrinkage to be higher in wild-type interfaces compared to *twist*, this difference is statistically not significantly different (Fig 4A). Because wild-type cell interfaces have been stretched by mesoderm invagination, they will generally be longer in length when they initiate junctional shrinkage for intercalation. Therefore, we also compared the speed of junctional shrinkage in microns per minute. We find that wild-type cell junctions initially shrink significantly more quickly than *twist* in absolute terms (S4E Fig). In summary, although wild-type interfaces start longer and shrink more quickly in absolute terms, in proportional terms, they shrink at approximately the same rate (no significant difference) but with a tendency for wild-type junctions to shrink slightly quicker.

Since Myosin II is proposed to be the main source of mechanical stress driving junction shrinkage [19,20], we next quantified the density of Myosin II specifically on shrinking interfaces (Fig 4B). We find that wild-type Myosin II densities tend to be higher than those of *twist* mutants, but with only a very brief period of statistically significant difference between the 2 genotypes, at 5 minutes prior to neighbour exchange. This is consistent with the very mild effect on Myosin II in response to mesoderm invagination described in the section above. The slight decrease in Myosin density on shrinking interfaces in the *twist* mutant compared to wild type could explain the slight decrease in their proportional contraction rate.

The proportional speed in junctional shortening should influence the rate at which cells exchange neighbours, and from above our prediction would be that there should be no difference between wild type and *twist*. We tracked discrete cell neighbour exchanges (T1 swaps) over time (Fig 4C) (see Methods). We find that the number of T1 swaps increase from the start of GBE until about 20 minutes and that wild-type and *twist* embryos show a comparable proportion of T1 swaps throughout the course of GBE, with no discernible delay in *twist* compared to wild type (Fig 4D).

In summary, we conclude that there is no significant effect of mesoderm invagination on the proportional rate of junction shrinkage. However, wild-type junctions start longer and shrink more quickly in absolute terms and therefore are not delayed in reaching the point of neighbour exchange compared to *twist* mutants. Consistent with these observations, we find no difference in the proportion of cells undergoing intercalation events between wild-type and *twist* mutants. Therefore, there is no significant positive or negative effect of mesoderm invagination on the process of junctional shortening leading to cell intercalation events. However, we do see a slightly (not significantly) higher proportional rate of junction shrinkage in wild type compared to *twist*, accompanied by a slightly increased Myosin II density on

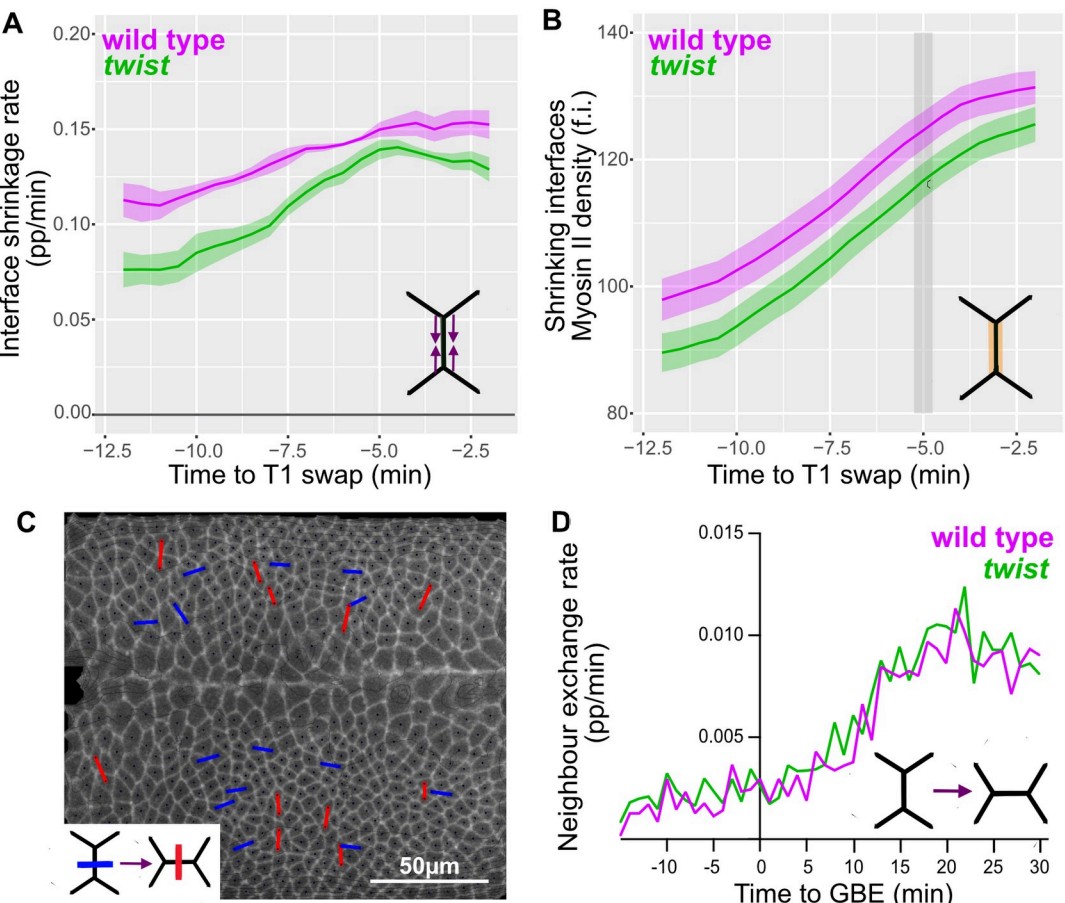

**Fig 4. Junctional shortening speed and rate of neighbour exchange in wild-type and *twist* mutants.** (**A**) Proportional rate of interface shrinkage plotted against time to T1 swap during GBE (data for 0–30 minutes of GBE) summarised for wild type (magenta) and *twist* (green). (**B**) Average density of Myosin II on shrinking interfaces, plotted against time to an intercalation event (swap) during GBE (data for 0–30 minutes of GBE) summarised for wild type (magenta) and *twist* (green). (**C**) Example of detection of neighbour exchange events in the germband in wild type at 20 minutes of GBE. Loss of neighbours is shown as blue lines between cell centroids, while gain of neighbours is shown in red. Cell centroids are shown as black dots. Underlaid is an image of Gap43Cherry signal at the level of adherens junctions extracted for tracking. (**D**) Rate of neighbour exchange. Number of interfaces involved in a T1 swap expressed as a proportion of the total number of DV-oriented interfaces for all tracked ectodermal germband cells, summarised for wild type (magenta) and *twist* (green) over time of GBE. Data associated with this figure can be found in S3 Data. DV, dorso-ventral; GBE, germband extension.

shrinking junctions. Therefore, junctional shrinkage appears to be mildly affected by a mechanosensitive response of Myosin II, and this may mask the effects of boundary conditions on this process.

## Comparing junctional elongation rate and AP cell elongation in wild-type and *twist* mutants

Following shrinkage of Myosin II−enriched cell junctions, cells exchange neighbours and cells previously separated from each other along the DV axis come into contact, forming a new junction (approximately AP-oriented) to drive extension of the tissue along the AP axis (Fig 3A) [9]. Cell junction elongation has been less studied than cell junction shrinkage in GBE. There might be active mechanisms involved, but biophysical modelling suggests that new

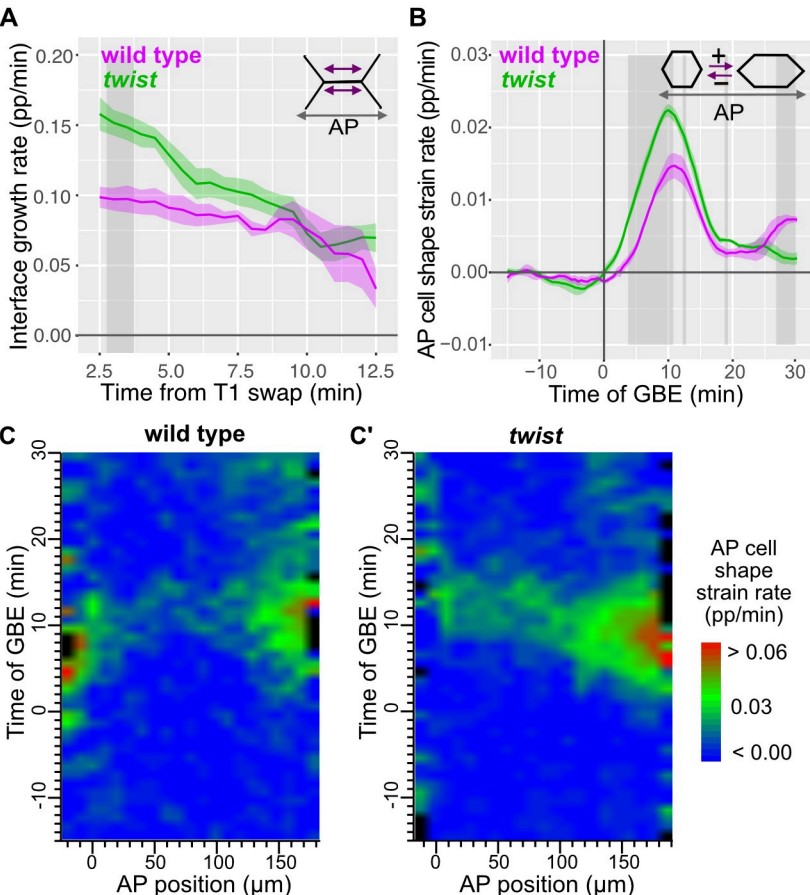

**Fig 5. Junctional elongation speed and AP cell elongation in wild type and *twist* mutants.** (**A**) Proportional rate of interface growth in ectodermal germband cells over time from the time of T1 swap summarised for wild type (magenta) and *twist* (green) (data for 0–20 minutes of GBE). (**B**) AP cell shape strain rate plotted against time of GBE summarising data from the ectoderm of the full imaged region of 4 wild-type and 6 *twist* movies. (**C, C'**) Spatiotemporal plots of AP cell shape strain rate plotted against AP position and time of GBE summarising wild-type (**C**) and *twist* (**C'**) movies. Data associated with this figure can be found in S4 Data. AP, antero-posterior; GBE, germband extension.

junction growth could also occur passively, following the resolution of unstable 4-way junctions [49]. Independently of the mechanism by which junctions elongate, active or passive, stress is expected to influence the dynamics of junctional elongation. For example, as for junctional shortening rates, the balance of boundary conditions on the extending ectoderm could impact junctional elongation rates. Indeed, while the posterior pull by endoderm invagination would promote junction extension, the counteracting orthogonal pull by mesoderm invagination would oppose junction extension. Therefore, an increase in junctional elongation rates might be expected in *twist* mutants, since new AP-oriented junctions would grow in the absence of an orthogonal tension (along DV) that could slow junction growth. The change in boundary conditions in *twist* mutants might also increase the AP cell shape strain rates since in absence of an orthogonal pull from mesoderm invagination, cells might elongate more due to the endoderm invagination pulling along AP.

We first quantified the proportional rate of elongation of new junctions in wild type and *twist* (Fig 5A). We restricted our analysis to 0 to 20 minutes of GBE, when the endoderm pull

along AP appears to have its strongest effect, based on the rate of AP cell elongation (measured as AP cell shape strain rate; [3,4] and see below). We found that new junctions grow significantly quicker in *twist* compared to wild type just after the T1 swap (Fig 5A). Then, junction elongation rates gradually become similar between the 2 genotypes. This suggests that in wild type, the DV tension in the germband that is caused by mesoderm invagination slows the initial rate of elongation of new junctions.

We next measured AP cell shape strain rate. Consistent with our previous studies, cells elongate along AP from the onset of GBE until about 20 minutes in wild type (Fig 5B), with a characteristic spatiotemporal gradient of elongation highest towards the posterior of the embryo (Fig 5C) [3,4]. In *twist* mutants, AP cell shape strain rates are significantly elevated compared to wild type (Fig 5B), with a more pronounced posterior spatiotemporal gradient (Fig 5C'). As expected, in both genotypes, AP cell elongation is accompanied by an increase in cell area, showing that the cell apices are under tension (S5A and S5A' Fig). In wild type, cells are stretched first along DV by mesoderm invagination and then along AP by endoderm invagination, causing a substantial apical cell area increase across the ectoderm (S5A Fig). In *twist* mutants, the loss of the pull from mesoderm invagination causes a strong diminution in this cell area increase (S5A' Fig). There might be a limit up to how much the cells can increase their apical area, which, in turn, might limit how cells will stretch along AP in wild type. In *twist*, the stretching along DV does not take place because of the loss of mesoderm invagination, and this might allow the cells to stretch more along AP in response to endoderm invagination, as we observe (Fig 5B and 5C').

In contrast to in DV, we do not have a precise registration of the position of the cells along the AP axis in our movies, since the field of view is positioned around the middle of the embryo, with no landmarks available for registration (see Methods). To address a possible impact of position variability along AP, we examined individual spatiotemporal plots for AP cell shape strain rates for each wild-type and *twist* movie (S5B and S5C Fig). We find that the variation of the anterior edge of the field of view means that in some plots but not others, some AP cell elongation signal contributed by the cephalic furrow is included (the cephalic furrow separates the head from the trunk of the embryo and forms just outside the anterior edge of the field of view). Also, the variation of the posterior edge of the field of view means that more or less of the posterior gradient is in view (this will depend on the centering of the field of view and variation in embryo shapes). To control for these variations, we selected a 100 μm-long central region on the plots, systematically excluding the signal at the anterior due to cephalic furrow formation (boxed regions in S5B and S5C Fig) (Methods). We find that AP cell elongation is also higher in this central region in *twist* mutants compared to wild type, with several bursts of significance (S5D Fig), consistent with our analysis with the entire field of view (Fig 5B).

In summary, we conclude that a change in the DV boundary conditions in *twist* mutants causes both an increase in the rate of elongation of new junctions following T1 transitions and an increase in AP cell elongation contributing to GBE. This is in contrast with our finding that the change in boundary conditions does not affect junction shrinkage.

## Comparing the orientation of cell interfaces during cell intercalation in wild-type and *twist* mutants

Next, we asked whether the observed changes in the alignment of cells and cell junctions with the embryonic axes, in *twist* compared to wild type, could impact tissue extension, since this could lead to polarised cell intercalation being less well aligned with the embryonic axes, which might lessen how effective convergence and extension is.

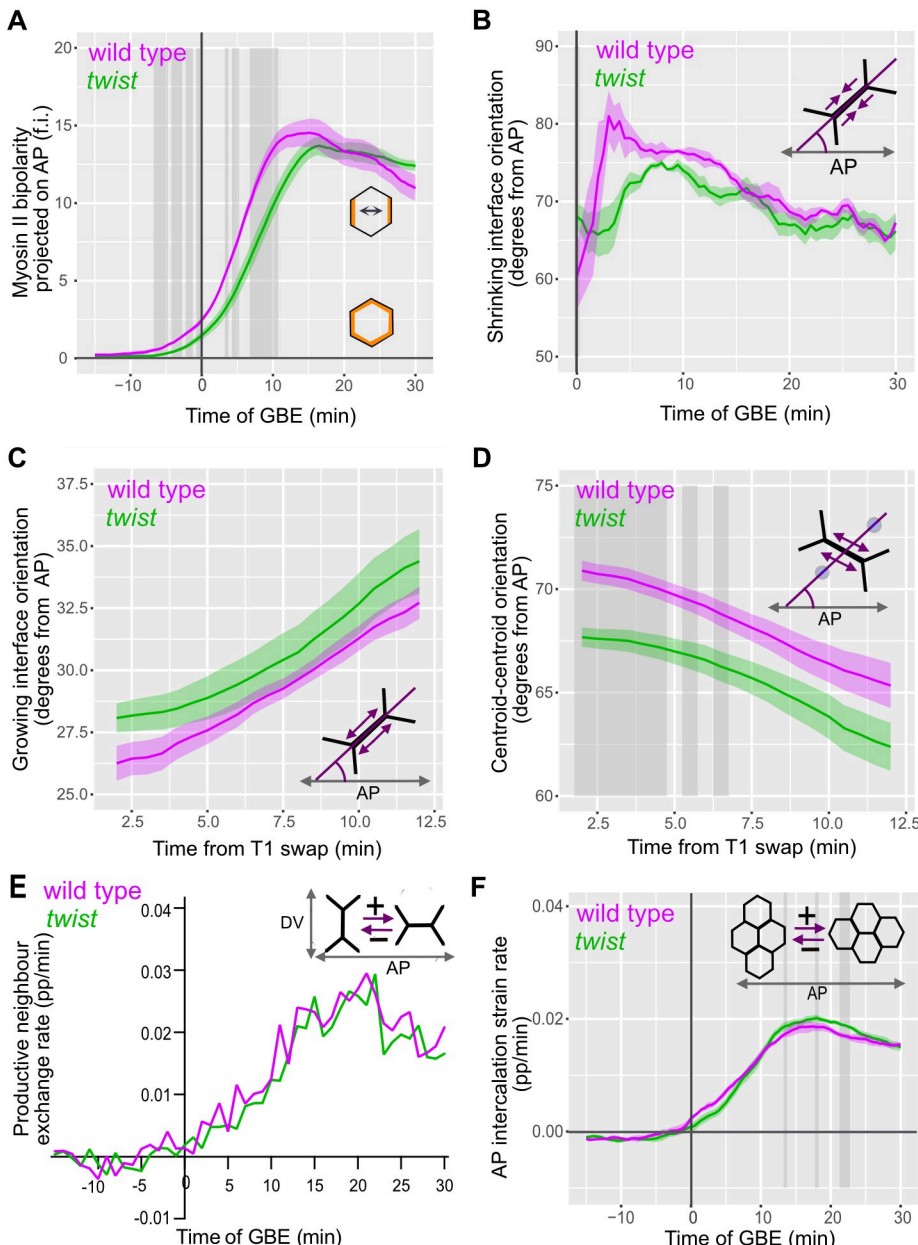

**Fig 6. Orientations of growing and shrinking junctions and their impact on AP intercalation strain rates in wild-type and *twist* mutants.** (**A**) Myosin II bipolarity, projected along AP, of all tracked ectodermal cells, summarised for wild type (magenta) and *twist* (green) plotted against time of GBE. (**B**) Orientation (degrees from AP axis) of shrinking interfaces at 6.5 minutes prior to T1 swap of all tracked ectodermal germband cells, summarised for wild type and *twist* over time of GBE. (**C**) Orientation (degrees from AP axis) of growing interfaces in ectodermal germband cells, summarised over the first 30 minutes of GBE for wild type and *twist* plotted against time from neighbour exchange event. (**D**) Orientation (degrees from AP axis) between cell centroids of pairs of "new neighbours" in ectodermal germband cells, summarised over the first 30 minutes of GBE for wild type and *twist*, plotted against time from neighbour exchange event. (**E**) Rate of productive neighbour exchange. Rate of net gains along the AP axis, using a continuous angular measure for productivity of T1 swaps to axis extension (see Methods), expressed as a proportion of the total number of DV-oriented interfaces for all tracked ectodermal cells, summarised for wild type (magenta) and *twist* (green) over time of GBE. (**F**) AP-projected intercalation strain rate of all tracked ectodermal cells, summarised for wild type and *twist* plotted against time of GBE. Data associated with this figure can be found in S5 Data. AP, antero-posterior; DV, dorso-ventral; GBE, germband extension.

First, we examined Myosin II bipolarity alignment with the embryonic axes in *twist* mutants compared to wild type. We projected our bipolarity cell measure along the anterior-posterior axis to assess the extent to which Myosin II is enriched at the anterior and posterior of each cell (Figs 6A, S6A and S6A'). We find that the rapid increase of AP-projected Myosin II polarity is delayed by approximately 5 minutes and significantly reduced in *twist* compared to wild type, until approximately 10 minutes into GBE. This difference is more pronounced than for the unprojected data (compare Fig 6A with Fig 3E), showing that the Myosin II bipolarity of cells is less well aligned with the embryonic axes early in GBE, in embryos where mesoderm invagination is defective. We conclude that the Myosin II–enriched junctions in *twist* mutants are less well aligned perpendicular to the AP axis than in wild type.

While the number of intercalation events is not reduced in *twist* compared to wild type (Fig 4D), the changes in average cell junction orientation (Fig 2K) and AP-projected bipolarity of Myosin II (Fig 6A) suggest that shrinking interfaces involved in T1 transitions may be poorly aligned with the DV axis. So, we measured the orientation of shrinking interfaces shortly before neighbour exchange and throughout GBE (Fig 6B) (Methods). We find no statistically significant difference between the orientations of shrinking interfaces in wild type and *twist*. In the first 5 minutes of GBE, there is a trend for shrinking interfaces to be less well aligned with the DV-axis in *twist* mutants, but this difference does not exceed 15 degrees (Fig 6B). These results suggest that the orientation of actively shrinking interfaces is not predominantly controlled by the global tissue movements and tension induced by mesoderm invagination. Because the maximum deviation in shrinking junctions misalignment we observe is 15 degrees at the start of GBE, then the *maximum* expected difference in strain rate projected along the DV axis from this misalignment would be: $1 - \cos(15) = 3.4\%$, predicting a very minor effect. Consistent with this, we found no significant reduction in DV intercalation rate contributing to convergence, in *twist* embryos compared to wild-type embryos (see S4B Fig, the negative values show the convergence rate in DV).

Previously, in other mutants affecting the orientations of shrinking interfaces, we had seen that the angle between junction shrinkage and growth was maintained (approximately 90 degrees), so growing junctions were also misoriented [50]. So, we also assessed the orientation of new junction growth. We did not detect a statistically significant difference in orientation of growing interfaces in *twist* mutants compared to wild type, although the trend is for *twist* to be less well aligned with the AP axis compared to wild type (Fig 6C). When plotting the angles between the centroids of newly neighbouring cells, we do see that these centroid–centroid angles are significantly less well aligned with the DV axis in *twist* mutants than in wild type for the first few minutes after a neighbour exchange (Fig 6D). Therefore, although T1 neighbour exchanges occur at normal rates in *twist* mutants (Fig 4D), the resulting cell packing is mildly perturbed compared to wild type soon after the neighbour exchange, but this effect is short-lived. We also quantified the rate of "productive" neighbour exchanges contributing to tissue extension (that is net T1 gains along the AP axis, taking into account the varying angles of T1 events with respect to the embryonic axes, see Methods) and observe that it is very similar in *twist* compared to wild type (Fig 6E). This shows that minor observed differences in angles of T1 events between wild type and *twist* do not impact on the ability of T1 events to extend the ectodermal germband.

Finally, we measured the intercalation strain rate in AP to ask whether the minor differences that we observe in junction growth speed and orientation cause any difference in the extension of the tissue by cell intercalation (Fig 6F). This is a continuous measure of how much cell intercalation is contributing to tissue extension in AP [3,40] and therefore will integrate effects on the numbers of T1 events, speed and orientation of junctional shrinkage, and junctional growth (see S6B Fig for the definition of intercalation strain rate). We find no

convincing difference between wild-type and *twist* embryos, though there is a trend for *twist* intercalation strain rate to be slightly higher, with several short bursts of significant differences between wild-type and *twist*. This tendency for *twist* intercalation strain rate to be mildly higher is consistent with our findings that the junctional growth rate is increased after T1 events (see Fig 5A).

In summary, although there are measurable differences in orientation of shrinking and growing interfaces, the effect on tissue extension is negligible. Considering all our results together, we conclude that the polarised cell intercalations causing GBE are not significantly augmented by mesoderm invagination and that they are robust to the extrinsic force that it exerts.

### Comparing tissue extension rate in wild type and *twist* mutants

Finally, we sought to compare how the different effects analysed above impact on total tissue extension during GBE. We used our measure of tissue strain rate along AP to address this [3,40]. We find that AP tissue strain rates are higher in *twist* mutants than in wild type, showing that germband extension is proceeding faster in *twist* mutants than in wild type (S6C Fig). This elevated rate of AP extension is expected from our observations that the AP cell elongation is higher in *twist* than in wild type, while there is little difference in the AP intercalation strain rate. This is because in our mathematical description, tissue strain rate is the sum of cell shape strain rate and intercalation strain rate (S6B Fig) [3,40]. Since AP cell shape strain rate varies along AP (S5B and S5C Fig), so will AP tissue strain rate. So as for the AP cell shape strain rate, we analysed AP tissue strain rate for a central 100 μm-wide region as before, to control for slight variability in the AP positioning of the field of view between movies. We confirm that in this central region, AP tissue strain rates are still significantly higher in *twist* mutants compared to wild type (S6D Fig), confirming results for the full field of view (S6C Fig).

## Discussion

We set out to investigate whether mesoderm invagination impacts on GBE in *Drosophila* embryos. We find that despite mesoderm invagination pulling and transiently deforming the adjacent ectoderm, the process of polarised cell intercalation is remarkably unaffected by this mechanical interaction, revealing its robustness to a counteracting tensile stress. In addition, we find that GBE is faster in absence of mesoderm invagination, because cell elongation along AP increases. We propose this is caused by the change in boundary conditions, with the effect of the posterior endoderm invagination pull on cell shapes becoming stronger in absence of mesoderm invagination. Our findings are summarised in Fig 7.

### Mechanical impact of mesoderm on the ectoderm

Previous studies looking at the mechanical impact of mesodermal cell contraction have focused on cell deformation *within* the mesoderm domain [35–38]. These studies have shown that while the ventral-most region of the mesoderm undergoes apical constriction, a region of about 3 cells on either side (the most lateral of those being the mesectoderm cells) do not constrict but instead stretch along DV. Beyond the mesectoderm cells, the lateral ectoderm cells were thought to be "inert" and mechanically buffered by the stretching of compliant tissue at its ventral (and also dorsal) edge [13,35,36]. In this paper, we show that ectodermal cells do respond mechanically to mesoderm invagination. We detect a ventral to dorsal gradient of cell stretching in the lateral ectoderm, which is lost in *twist* mutants (Fig 2G). This suggests that forces generated by the contractile activity of the apical actomyosin in the mesoderm do transmit to the ectoderm via the apical adherens junctions. When mesoderm invagination begins,

**with DV pull (wild type)**                    **no DV pull (*twist*)**

Myosin II behaviour

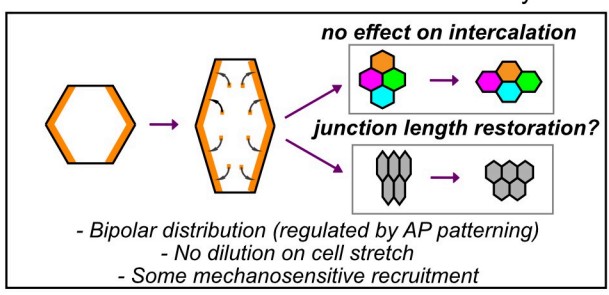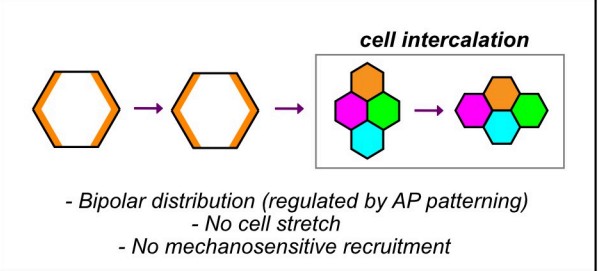

Mechanical force balance

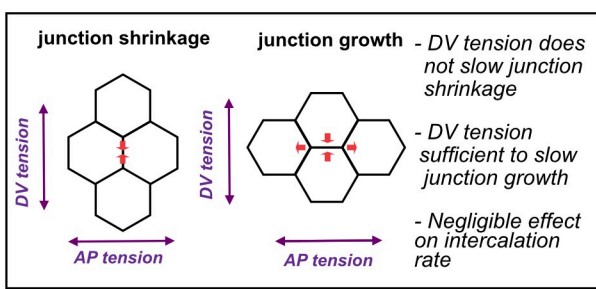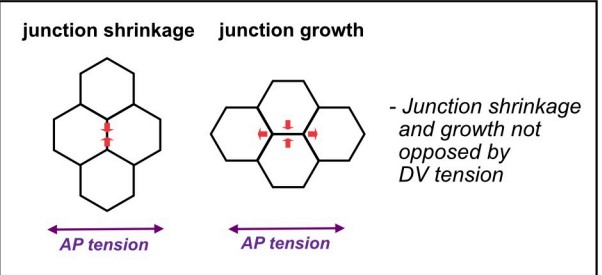

Cell orientations

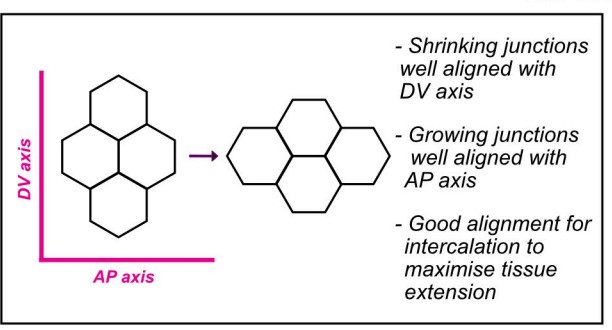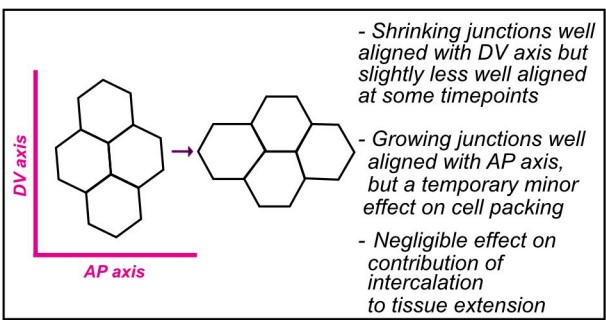

**Fig 7. Summary of the consequences of the mesodermal pull on cell and junction behaviours associated with polarised cell intercalation during GBE.** Left column considers the behaviours in the presence of the mesodermal pull (in wild-type embryos), and right column considers these in the absence of the mesodermal pull (in *twist* mutant embryos).

the level of apical Myosin II is still very low in the lateral ectodermal cells (Fig 3D), which might explain why ectodermal cells stretch significantly under tension from the invaginating mesoderm, relieving the mechanical stress. We also show that ectodermal cell stretching is not accompanied by cell slippage.

## Mechanisms regulating Myosin II density at the cortex and consequences for polarised cell intercalation

The stretching of ectodermal cell–cell junctions could trigger a mechanosensitive response leading to an increase in Myosin II enrichment, or, alternatively, a dilution of Myosin II (Fig 7). Although we do not find evidence of a significant increase in Myosin II density (see below), we find a clear absence of dilution. This suggests that a homeostatic mechanism maintains

Myosin II density at the cortex upon rapid stretching. It might be that the AP patterning system controlling Myosin II enrichment at DV-oriented interfaces senses the density of Myosin II at the cortex and maintains it at a specific density. Alternatively, the homeostatic mechanism might be a more general mechanosensitive response to stretch. One consequence of maintaining Myosin II density in stretching DV-oriented junctions is that these junctions undergo shrinkage to initiate neighbour exchange events in wild-type embryos, at the same rate as unstretched junctions in *twist* mutants. In contrast, in the case of the lateral mesoderm, the already low levels of apical medial Myosin II become further diluted and are easily overcome by the pulling forces of the central mesoderm. This causes an extreme apical stretching of cells, rather than homeostatic maintenance of apical actomyosin density [35]. This highlights the fact that responses of cells to apical tension is cell-type specific.

We did find that Myosin II recruitment rate was temporally marginally higher in wild type than in *twist*, providing further evidence of a mechanotransductive response. It is possible that the homeostatic mechanism discussed above is triggered by mechanical stretch and overshoots slightly, leading to a marginally increased rate of Myosin II recruitment. However, this increase is at the limit of detection and did not have a significant effect on the proportional shrinkage of cell junctions during neighbour exchange.

Mechanosensitive feedback in Myosin II recruitment has been reported at cable-like actomyosin enrichments spanning several junctions connected to each other later in GBE and at parasegmental boundaries, due to linked junctions pulling on each other [24,48]. In both cases, when the "cables" were cut by laser dissection to decrease tension in the linked junctions, a modest decrease in Myosin II recruitment was measured (less than 10%). Considering that mesoderm invagination pulls on the ectoderm when apical actomyosin is very low and just starting to increase (Fig 3D), it is perhaps unsurprising that any mechanosensitive increase in Myosin II would be challenging to detect.

## Comparison with other studies

Our results are in contrast to some other studies that have reported a stronger effect of mesoderm invagination on Myosin II localisation and GBE, but there are inconsistencies in results between studies, perhaps due to the challenge of acquiring comparable datasets between wild type and mutants, so the precise effect of mesoderm invagination on GBE has not been clear [3,4,27–30]. Indeed, the first study noting a possible reduction of cell intercalation in mutant embryos lacking mesoderm invagination was from our own laboratory [3]. Since this study, we have improved cell tracking techniques (for example, manual correction of mistracked cells and more precise exclusion of the mesoderm/mesectoderm cells) and started imaging earlier which has improved movie synchronisation and allowed us to collect more cells for analysis around the time of mesoderm invagination (see Methods). Additionally, there are now more fluorescently tagged proteins available, and we have taken care here to choose markers that are expressed at the same level between wild-type and *twist* embryos. For these reasons, along with the consistency between our different analyses, we are confident in the results that we present here.

## Restoration of DV cell length after being stretched by mesoderm invagination

Following mesoderm-induced DV cell and cell junction stretching, cells contract to regain an approximately isotropic shape by the end of GBE [3]. When comparing junctional shrinkage leading to neighbour exchange, we found that the proportional rate of length change was not significantly different, but in absolute terms (μm/minute), wild-type junctions shrink faster

than those in *twist* mutants. This is because wild-type DV-oriented junctions have been stretched, and these cell junctions are both restoring their length after stretching and undergoing shrinkage for intercalation events simultaneously. The transient marginal increase in the rate of Myosin II recruitment in wild type compared to *twist* (Fig 3) may be due to a mechano-transductive mechanism that acts to restores junctional length after stretching. Other mechanisms might also contribute to the restoration of cell shape after stretching, including the material properties of the cells, the activity of medial myosin networks at the apices of cells, and three-dimensional constraints to their shape (the volume of cells is thought not to change in the early embryo [51]). Nevertheless, the possibility remains that mechanosensitive recruitment or stabilisation of Myosin II on the cortex of stretched junctions is important for cell junctions to resist overstretching and subsequently restore their length. In the context of GBE, however, this does not significantly help to drive junctional shortening for cell intercalation as we have shown by comparing proportional junctional shrinkage rates.

## Boundary conditions during germband extension and impact on junction shortening and elongating rates

Loss of mesoderm invagination would be expected to change the boundary conditions during GBE. The extending germband is pulled along AP as a consequence of posterior endoderm invagination, while mesoderm invagination pulls along DV. These 2 pulls can be considered as boundary conditions acting in perpendicular directions on the extending germband. Even after mesoderm invagination is complete, the DV tension may remain as the mesectodermal cells meet and adhere to each other at the midline, which could hold tension in the ectoderm. This DV-oriented force could counteract both the AP pulling force and forces generated within the germband, promoting convergence and extension, such as those generated by Myosin II contraction at shrinking interfaces. If so, the rate of junction shortening and elongation would increase in the absence of mesoderm invagination. We find a clear increase in the initial rate of junction growth, and in cell elongation along AP, in *twist* mutants (Fig 5). This suggests that mesoderm invagination does act as a counteracting force. Interesting, following the initial dissipation of stress from resolving the unstable 4-way vertex, the rate of junctional elongation slows and become comparable in wild-type and *twist* mutants, suggesting that it is the initial phase of junctional growth that is affected by the global force balance of boundary conditions. We note that despite the increase in initial junctional elongation rate seen in *twist* mutants, we see little impact of this on the overall rate of cell intercalation.

We do not, however, detect a similar increase in the rate of junctional shortening in *twist* mutants (in fact, it tends towards a reduction, Fig 4). Based on the above, it is plausible that an increase in rate of junctional shortening due to mechanical effects might be cancelled out by a decrease in rate of junctional shortening linked to a decrease in Myosin II mechanosensitive recruitment in absence of cell stretching. In this model, the opposing force of mesoderm invagination to ectoderm extension would be balanced by the increased contractility of the stretched interfaces.

## Junction orientation and tissue extension

We addressed whether mesoderm invagination impact ectoderm extension by helping to align the junctions of rearranging cells along the main embryonic axes. We do find a significant effect of mesoderm invagination on cell and junction orientation up to approximately 15 minutes of GBE. However, the effect on the shortening cell junctions themselves is marginal, even within this first 15 minutes of GBE, suggesting that their orientation is relatively insensitive to tissue-level forces. This could be explained by the rapidly increasing Myosin II planar

polarisation of the ectoderm during this period. We and others have shown that Myosin II–enriched interfaces form supracellular cable-like structures that straighten throughout the course of GBE [24,26,47]. Thus, this tissue-autonomous straightening might explain why the orientation of shortening junctions is mostly unaffected by mesoderm invagination. Consistent with this, [24] report that these cable-like structures appear normal in *twist snail* double mutants later in GBE.

We also found only marginal differences in the orientation of growing junctions between wild-type and *twist* mutants. This is expected since junctions grow in an orientation perpendicular to shortening junctions, and this angle is tightly constrained [50]. Also, invagination of the posterior endoderm (which still occurs in *twist* mutants) will act to pull growing junctions to be parallel with the AP axis [9]. Overall, we find that the minor differences we observe in the orientation of shrinking and growing interfaces do not have an impact on the productivity of cell intercalation events for tissue extension.

## Methods

### Fly maintenance and strains

We used a CRISPR knock-in line in the *spaghetti-squash* locus (sqh-eGFP[29B], referred to in this work as sqhGFP[KI]) to tag endogenous Myosin II regulatory light chain (MRLC) with GFP [39]. The line is homozygous viable and so all the molecules of MRLC are labelled with GFP. We used Gap43mCherry (on the X) to mark cell membranes [52]. A recombinant chromosome between sqhGFP[29B] and Gap43Cherry was made using standard *Drosophila* genetics. We used the null mutation *twi[1]* (Bloomington Stock Center). *twist* homozygous embryos were collected from sqhGFP[29B], Gap43Cherry; *twi[1]*/Cyo stock and wild type from sqhGFP[29B], Gap43Cherry; Gla/Cyo stock. Standard fly husbandry was used to maintain stocks.

### In situ hybridisation chain reaction and antibody staining

In situ hybridisation chain reaction (HCR) and immunostaining methods are as described previously [47]. Stage 5 to 8 embryos were collected from *twi[1]*/CyO stocks. Morphology of the ventral furrow and ectopic expression of *ind* or *sim* within the mesoderm was used to identify mutants. The HCR probe sets were designed by Molecular Instruments. Primary antibodies used were mouse anti-phospho-Tyrosine (pTyr) (Cell Signalling #941) (1:1,000) and rabbit anti-phospho-Histone3 (pHis) (Cell Signalling #9701; 1:200). Secondary antibodies conjugated to fluorescent dyes were obtained from Jackson ImmunoResearch Laboratories, Invitrogen, and Life Technologies. Streptavidin with Alexa Fluor 405 conjugate was from Thermo Fisher Scientific.

Embryos were imaged on an inverted SP8 Confocal Microscope (Leica Systems), with a 40X 1.3NA oil-immersion objective. Either a PMT or an HyD detector was used alongside a 405/488/546/647nm laser line. Image stacks of 1 μm Z separations were captured using the Leica Application Suite X Software.

### Movie acquisition

Embryos were mounted ventral/ventrolaterally and a central region of the embryo was imaged on a spinning disc confocal microscope as previously [26]. The anterior edge of the field of view was typically just posterior to the cephalic furrow separating the head from the trunk of the embryo (Fig 1F). Z-stacks, with 1 μm step size, were collected sequentially for each channel. Embryos were imaged every 30 seconds from approximately mid-cellularisation (stage 5) until the onset of cell divisions within the field of view (approximately stage 8). We imaged at a

controlled temperature (20.5˚C +/− 1˚C) to minimise variability, since *Drosophila* developmental processes are temperature dependent [53,54].

We checked that wild-type embryos survived to larval hatching post acquisition. *twi* homozygous embryos were identified by failure of ventral furrow formation during movie acquisition and then left to develop until the end of embryogenesis to confirm their phenotype by cuticle preparation (*twi* is an embryonic lethal gene so homozygous embryos die before larval hatching).

## Movie tracking

To track at comparable apico-basal positions in all cells regardless of embryo curvature, acquired z-stacks were converted into stacks of curved quasi-two-dimensional representations, following the curved surface of the embryo at each time point. Next, automated segmentation, with manual intervention to increase tracking accuracy, was performed as previously [26]. We segmented cell contours in the Gap43Cherry channel at the level of adherens junctions (see example S1 and S2 Movies). During GBE, we used the Myosin II signal concentrated at the level of adherens junctions as a guide to ensure we were segmenting at the correct plane. Note that prior to GBE, apical Myosin II is very weak and apical adherens junctions are slightly more basal, based on the ruffled appearance of cell membranes more apically and imaging data of Cadherin-GFP in other datasets from our laboratory. Therefore, in earlier time frames, we used images from up to 4 μm deeper into the embryo to accurately segment cell contours.

Any inaccurately tracked cells were removed prior to analysis based on cell velocity relative to neighbours, cell area, cell area change, and number of frames over which the cell could be tracked.

The ventral midline (middle of the presumptive mesoderm in *twist)* was followed over time and used to define the orientation of the AP axis, and the DV axis was set perpendicular to this. "Distance from mesectoderm" is defined as the distance from the first row of centroids of ectodermal cells at 30 minutes of GBE.

## Note on number of cells in movies

In wild-type movies, initially, the mesoderm takes up a large part of the field of view leaving ectoderm only visible on one side of the embryos at the start of the movies (see example S3 Movie). As the mesoderm invaginates, the ectoderm on the other side of the embryo comes into view, and the number of ectodermal cells visible continues to increase as mesoderm invagination completes and ectodermal cells converge towards the midline during GBE. Note that the ectoderm that is visible often moves slightly dorsalwards and out of the field of view prior to mesoderm invagination, accounting for the reduction in cell numbers between −15 and 0 minutes of GBE (S3D and S3E Fig).

In *twist* movies, cell numbers vary less throughout the course of the movie because the mesoderm primordia is narrower in *twist* than in wild-type so takes up less space initially but does not invaginate successfully so still takes up part of the field of view throughout GBE (see example S4 Movie and S3F Fig). Precise cell numbers vary slightly between movies due to differences in exact DV location of mounting of the embryo and slight variations in the number of cells successfully tracked.

## Movie synchronisation

Movies of the same genotype were synchronised together using a threshold of 0.01 pp/minute AP-projected tissue strain rate as previously [3]. This nonzero threshold was chosen due to some minor fluctuations in this strain rate around the start of GBE. Average AP-projected

tissue strain rate curves were then calculated for each genotype, and used to synchronise movies so that average AP-projected tissue strain rate was 0 pp/minute at time = 0 minutes GBE (S3A and S3B Fig).

For the purposes of checking synchronisation, the appearance of apical myosin in the ectoderm was assessed by eye as signal discernible above background in raw movies (S3C Fig). Cytokinesis timing was assessed through by-eye observation of Myosin II–enriched cytokinesis rings.

## Exclusion of mesoderm and mesectodermal cells from analysis

Cell types were initially broadly defined using rules based on their proximity to the midline at a given time point and then were refined by manually selecting cells of incorrect cell-types based on their patterns of cell division within the time-period of analysis and redefining them where necessary. Note that in wild type, the width of the mesoderm is approximately 18 cells, while in *twist* mutants, the width of the mesoderm is reduced to approximately 8 to 10 cells, and this was taken into account when defining initial midline proximity rules [41] (see examples S3 and S4 Movies).

For cell division timing, we used comparison to fixed embryos with mesectodermal (*sim* HCR) and dividing cells (anti-phospho-H3) marked to ensure accurate cell type identification throughout the analysed time-period (S1A and S1B Fig). The pattern of cell division was of particular use for the *twist* movies because the boundary between ectoderm and mesectodermal cells is not as straight as in wild type (S1B–S1B" Fig).

## Selection of a 100 μm-wide region to control for variations in AP registration

In contrast to DV, our movies are not registered precisely along the AP axis because of lack of landmarks. To control for variations in AP registration, we compared both the imaged ectoderm for the full field of view and also for a smaller 100 μm "central" region along AP (S5B and S5C Fig). The anterior boundary of the central window was chosen to exclude cells that underwent higher AP cell shape change compared to elsewhere in the field of view, due to the proximity to the cephalic furrow (which typically formed just outside the anterior edge of the field of view) (white dotted line in S5B and S5C Fig). The posterior boundary was positioned 100 μm further along AP, thus defining a central region. This central region was used for quantitative comparison of AP cell shape (Figs 5B and S5D) and AP tissue strain rates (S6C and S6D Fig) between wild type and *twist*.

## Movie analysis

**Analysis of cell and cell junction behaviour.** Cell shape strain rates were calculated using minimisation to find the best mapping of cell shapes in adjacent pairs of movie frames. Then, tissue, cell shape, cell intercalation, and cell area strain rates were calculated as previously [3,4]. Axial shape elongation was calculated as previously [3,50]. Straight vertex to vertex interface lengths and orientations were calculated as previously [50]. Cell orientation is the orientation of the long axis of the cell relative to the AP embryonic axis. Rates of change are calculated from 2 time points (1 minute) before, to 2 time points after the time point in question.

Neighbour exchange and productive neighbour exchange measures were calculated as previously [50], except with a modified rule for whether a T1 was productive (extending in AP) or counterproductive (extending in DV). Previously, T1s were defined as productive when the angle of the centroid–centroid line between the pair of cells that make a new contact was greater than 45 degrees from AP. Here, rather than discrete, we made each T1 contribution

continuous, depending on the angle of the centroid–centroid line because we observed that the orientation of *twist* T1s were less strictly aligned with embryonic axes than in the wild-type. T1 contributions ranged continuously from 1 for a centroid–centroid line aligned with DV, through 0 at 45 degrees, to −1 if aligned with AP. Both neighbour exchange measures were expressed as a proportion of the total number of DV-oriented interfaces and as a rate per minute so that they were directly comparable to our intercalation strain rates in units of proportion per minute.

**Myosin II quantification.** We quantified junctional Myosin II as it is the driver of junctional shrinkage. Medial Myosin II has also been implicated in cell intercalation and has been shown to flow towards shrinking junctions helping to drive their Myosin II enrichment [25]. Therefore, junctional Myosin II can be taken as a readout of polarised Myosin II behaviour.

Fluorescence quantification was performed on raw data from the Myosin II channel that had been corrected for artefactual domed Myosin intensity across the field of view as previously [26] and normalised as described in the section below.

Tracking of the Gap43 channel was used to define positions of cell interfaces. Because channels were imaged sequentially, images are slightly out of register with each other due to the time elapsed between capturing each channel. Therefore, channel registration was corrected post-acquisition in order that information on the position of interfaces in the Gap43 channel could be used to locate them in the Myosin channel. Therefore, the local flow of cell centroids between successive pairs of time frames in the Gap43 channel is used to give each interface/ vertex pixel a predicted flow between frames. A fraction of this flow is applied, equal to the Myosin II to Gap43 channel time offset, divided by the frame interval. Because cells deform as well as flow, the focal cell's cell shape strain rate is also applied, in the same fractional manner as above. Success of this channel registration was confirmed by overlaying the final result of quantification after registration on top of the Myosin channel images. Examples of this can be seen in Fig 3B' and 3C'.

We quantified Myosin II associated with cell junctions from maximum intensity projections of 5 z-planes from quasi-two-dimensional stacks extracted prior to tracking (see Movie tracking) from 2 μm above to 2 μm below the level of the adherens junctions during GBE. These thicknesses and projection depths were chosen to capture the majority of junctional-associated myosin while minimising contamination with nonjunctional signal.

Cell junctions/interfaces were defined as a 1-pixel thick line surrounding each cell, with values for shared interfaces averaged between these two 1-pixel thick lines. Because cell vertices are often enriched in Myosin, vertex pixels (within a radius of 2 pixels of the vertex point) were removed for the quantification.

We then calculated the bipolarity (planar cell polarity, with 2 peaks approximately at opposite cell edges) of each cells average interface myosin by Fourier analysis, extracting the average amplitude of 2 Myosin peaks when plotting the interface fluorescence intensity around the cell perimeter (see also [26]). Prior to Fourier analysis, elongated cells were unstretched to eliminate bias introduced by unequal interface lengths. The phase of the period 2 component of the Fourier series gives the direction of the bipolarity. The bipolarity is presented as both unprojected values and projected to the embryonic anterior-posterior axis. Rates of change are calculated from 2 time points before to 2 time points after the time point in question.

## Normalisation of Myosin channel

Background signal was subtracted by setting pixels of intensities up to fifth percentile set to zero for each time point. The fifth percentile was chosen as this removed background signal without removing any structured (i.e., nonuniform) Myosin signal, as assessed by eye.

Intensities varied slightly between experiments due to differences in laser intensity, and, therefore, histograms of pixel intensities were stretched: The 98.5 percentile of the data at 30 minutes GBE was set to a greyscale value of 200 with lower value pixels scaled by the same factor and leaving the top 55 values for a tail of high value pixels. We choose to stretch histograms using a reference time point, rather than on a per time point basis, to preserve the observed increase in Myosin intensity over time. Examples images and histograms of movie frames before and after normalisation can be seen in S7A–S7D' Fig.

Following normalisation, we quantified "nonjunctional" Myosin II from the same apicobasal position in the cell (+/− 2 μm from adherens junctions) as we quantify junctional Myosin II (see above).

Note that "nonjunctional" Myosin II is defined as the remaining pixels in the cell after the definition of junctional and vertex pools of Myosin II (see details in Myosin quantification section above) and will include both cytoplasmic signal and some medial web associated Myosin II. We compared the "nonjunctional" Myosin II intensities between wild type and *twist* (see S7E Fig) and find that they are very similar, confirming that that we are not inappropriately brightening one set of movies compared to the other.

### Statistics and graphical representations

Ribbon plots were generated in R with 30-second bins and profiles smoothed over 3 bins for presentation only (significance tests were calculated on unsmoothed data). We used a mixed effects model to test for differences between genotypes with a *p*-value of 0.01 as previously [50]. This estimates the *p*-value associated with a fixed effect of differences between genotypes, allowing for random effects contributed by differences between embryos within a given genotype. Significant differences are highlighted in grey on plots. Ribbons show standard error between genotypes. Spatiotemporal plots were generated in custom software otracks [40]. Other plots (neighbour exchanges, synchronisation plots) were generated in Prism with 1-minute bins.

### Supporting information

**S1 Fig. Defining cell types in wild-type and *twist* mutant movies.** (**A**-**B**") Representative images of in situ HCR of *sim* (green, mesectoderm) with antibody staining of pH3 (magenta, dividing cells) in stage 6 to 8 fixed embryos for wild type (**A**-**A**") and *twist* (**B**-**B**"), used to inform definition of cell types in movies. (**C**, **D**) Examples of defining cell types in wild-type (**C**) and *twist* (**D**) movies at representative time points (mesoderm/mesectoderm, cyan; ectodermal germband, magenta), overlaid on Myosin II channel (before normalisation, maximum intensity projection). Unmarked cells are poorly tracked and excluded from the analysis. See also S3 and S4 Movies. HCR, hybridisation chain reaction; pH3, phosphoHistone3; *sim*, *single-minded*.
(TIFF)

**S2 Fig. Expression of DV patterning genes and LRR receptors in wild-type and *twist* mutants.** (**A**-**D**) In situ HCR of DV patterning genes in wild-type and *twist* mutant (**A**'-**D**') stage 7 embryos: (**A** and **A**') *sim* (magenta) and *vnd* (green); (**B** and **B**') *ind* (magenta) and *vnd* (green); (**C** and **C**') *twist* (green), which is missing in *twist* mutants (**D** and **D**') *snail* (green), which is reduced and patchy in *twist* mutants as expected. (**A**-**D**') are also marked with anti-pTyr (white, cell membranes). HCR of LRR receptors (green) in wild-type (**E**-**H**) and *twist* mutant (**E**'-**H**') stage 7 embryos: (**E** and **E**') expression of *Toll-2*; (**F** and **F**') expression of *Toll-6*; (**G** and **G**') expression of *Toll-8*; (**H** and **H**') expression of *trn*. Maximum intensity

projections of approximately 20 μm image stacks. (**E-H'**) are also stained for *ind* expression (HCR, magenta) and marked with pH3 (blue, dividing cells). DV, dorso-ventral; HCR, hybridisation chain reaction; *ind*, *intermediate neuroblast defective*; LRR, Leucine-rich repeat; pH3, phosphoHistone3; pTyr, phosphoTyrosine; *sim*, *single-minded*; *trn*, *tartan*; *vnd*, *ventral nervous system defective*.
(TIFF)

**S3 Fig. Numbers of cells analysed in wild-type and *twist* mutant movies.** (**A**, **B**) Synchronisation of movies to start of GBE using tissue strain rates (proportion/minute), projected along AP axis in wild type (**A**) and *twist* (**B**). (**C**) Movie synchronisation was checked by mapping by eye using key developmental events visible in our field of view. Events in the ectoderm ("ectodermal apical Myosin II" and "first lateral cytokinesis" are clustered together within same genotype and between genotypes, showing that movie synchronisation is good throughout our window of study (−15 to +30 minutes of GBE). Note that the movies are longer than the −15 to 30 minutes of GBE analysed and vary slightly in length. Also, data are missing for "first lateral cytokinesis" for *wt200910*, due to the movie finishing too early for this to be assessed. Lines show average time of developmental events (magenta, wild type; green, *twist*). (**D-F**) Number of ectodermal cells analysed in wild-type and *twist* mutant movies over time. (**D**) Total number of cells analysed in all movies; (**E**) number of ectodermal cells analysed per wild-type movie over time; and (**F**) number of ectodermal cells analysed per *twist* movie over time. Data associated with this figure can be found in S6 Data. AP, antero-posterior; GBE, germband extension.
(TIFF)

**S4 Fig.** (**A**) Assessment by eye of timings of 4 events during mesoderm invagination in wild-type movies: when the first sustained apical constrictions begin; when synchronous apical constrictions along the AP axis become evident; when the central mesoderm has buckled and is surrounded by 2–3 rows of stretched mesoderm and mesectoderm cells; and when mesoderm cells are all internalised, with the mesectoderm cell row still stretched. Times show minutes of GBE. Averages of first and last events (about −13 and + 7 minutes) were used to draw "mesoderm invagination" bars in Fig 2C and 2E. (**B**) DV-projected intercalation strain rate of all tracked ectodermal cells, summarised for wild type (magenta) and *twist* (green) over time of GBE. Note that positive values would show intercalation contributing to extension in DV. Negative values show convergence in DV, with more negative values showing stronger convergence. For completion, we are also showing in (**C**), the DV-projected tissue strain rate of all tracked ectodermal cells, summarised for wild type (magenta) and *twist* (green) over time of GBE. Note that intercalation strain rate is calculated as tissue strain rate minus cell shape strain rate [40] (see S6B Fig). Therefore, DV tissue strain rate in C is very similar to DV cell shape strain rate (Fig 2C) before the start of GBE as DV cell intercalation strain rates are close to zero between −15 and 0 minutes of GBE. (**D** and **D'**) Spatiotemporal plots of unprojected Myosin II bipolarity of all tracked ectodermal cells, summarised for wild type (**D**) and *twist* (**D'**) against time of GBE and distance from mesectoderm (μm) (see Methods). (**E**) Speed of interface shrinkage (μm/minute) plotted against time to T1 swap during GBE (data for 0–30 minutes of GBE) summarised for wild type (magenta) and *twist* (green). Data associated with this figure can be found in S7 Data. AP, antero-posterior; DV, dorso-ventral; GBE, germband extension.
(TIFF)

**S5 Fig. Cell area and AP cell elongation changes in wild-type and in *twist* mutants.** (**A**, **A'**) Spatiotemporal plots of cell area strain rate plotted against anterior-posterior position and time of GBE summarising 4 wild-type (**A**) and 6 *twist* (**A'**) movies. (**B**) Spatiotemporal plots of

AP cell shape strain rate plotted against AP position and time of GBE for each wild-type movie. (**C**) Spatiotemporal plots of AP cell shape strain rate plotted against AP position and time of GBE for each *twist* movie. Vertical dotted white lines on each plot in (**C**) and (**D**) show the anterior and posterior limits of the 100 μm-wide central region used to plot data in D (see Methods). (**D**) AP cell shape change strain rate plotted against time of GBE summarising data from the central region shown in (**B**) and (**C**) for wild-type and *twist* movies. Data associated with this figure can be found in S8 Data. AP, antero-posterior; GBE, germband extension. (TIFF)

**S6 Fig.** (**A**, **A'**) Myosin bipolarity projected along AP for all tracked ectodermal cells, summarised for wild type (**A**) and *twist* (**A'**) against time of GBE and distance from mesectoderm (μm) (see Methods). (**B**) Diagram illustrating that tissue strain rates are the sum of cell shape strain rates and cell intercalation strain rate [40]. Tissue and cell shape strain rates are measured and intercalation strain rates are deduced from those. (**C**) AP tissue strain rate plotted against time of GBE, summarising data from the full imaged region of 4 wild-type and 6 *twist* movies. (**D**) AP tissue strain rate plotted time of GBE summarising data from the 100-μm central region (see Methods) of wild-type (magenta) and *twist* (green) movies (see also S5 Fig). Data associated with this figure can be found in S9 Data. AP, antero-posterior; GBE, germband extension. (TIFF)

**S7 Fig. Normalisation of Myosin II channel.** Images of the Myosin II channel from movies prior to (**A-D**) and after (**A'-D'**) normalisation, for example, wild-type (200910_3) (**A-B'**) and *twist* (201119_1) (**C-D'**) movies. The normalisation method used was designed to enable meaningful comparison of Myosin II intensities between movies (compare **A'** to **C'** and **B'** to **D'**) while maintaining the observed differences in Myosin II intensity over time (compare 5-minute and 30-minute time points). See Methods for details. (**E**) Comparison of average "nonmembrane" apical Myosin II intensity in the ectoderm of 4 wild-type (magenta) and 6 *twist* (green) movies (see Methods). Fainter lines show standard deviations. Data associated with this figure can be found in S10 Data. (TIFF)

**S1 Movie. Example of cell tracking in a wild-type movie (prior to exclusion of mesoderm/mesectoderm).** Example wild-type movie (wt201118) from −15 to 30 minutes of GBE, showing manually corrected tracked cells with additional quality control rules applied to remove inaccurately tracked cells and incomplete cells at the edge of the embryo (see Methods). Cell contours (magenta) and centroids (white) overlayed on Gap43Cherry signal from the apical adherens junction level of the embryo, which was extracted by the "blanketing" of image stacks (see Methods; *Movie Tracking*). (MP4)

**S2 Movie. Example of cell tracking in a *twist* movie (prior to exclusion of presumptive mesoderm/mesectoderm).** Example *twist* movie (twi201118_1) from −15 to 30 minutes of GBE, showing manually corrected tracked cells with additional quality control rules applied to remove inaccurately tracked cells and incomplete cells at the edge of the embryo (see Methods). Cell contours (magenta) and centroids (white) overlayed on Gap43Cherry signal from the apical adherens junction level of the embryo, which was extracted by the "blanketing" of image stacks (see Methods; *Movie Tracking*). (MP4)

**S3 Movie. Example of cell types in a wild-type movie.** Example of defined cell types (mesoderm/mesectoderm, cyan; ectodermal germband, magenta) in wild-type movie (wt201118) from −15 to 30 minutes of GBE, overlayed on Myosin II channel (before normalisation, maximum intensity projection). Unmarked cells are poorly tracked and excluded from the analysis. (MP4)

**S4 Movie. Example of cell types in a *twist* movie.** Example of defined cell types (mesoderm/mesectoderm, cyan; ectodermal germband, magenta) in *twist* movie (twi201118_1) from −15 to 30 minutes of GBE, overlayed on Myosin II channel (before normalisation, maximum intensity projection). Unmarked cells are poorly tracked and excluded from the analysis. (MP4)

**S1 Data. Excel file for graphs in Fig 2.** Data for each graph (C, D, D', E, F, F', G, H, J, K) is given in a separate sheet containing raw data values of *x* and *y* axes, along with the corresponding cell and embryo identifier for each data point when appropriate (cell identifier and embryo name, respectively). Each sheet is labelled with the relevant figure panel number. (XLSB)

**S2 Data. Excel file for graphs in Fig 3.** Data for each graph (D, E, F, G, H) is given in a separate sheet containing raw data values of *x* and *y* axes, along with the corresponding cell and embryo identifier for each data point when appropriate (cell identifier and embryo name, respectively). Each sheet is labelled with the relevant figure panel number. (XLSB)

**S3 Data. Excel file for graphs in Fig 4.** Data for each graph (A, B, D) is given in a separate sheet containing raw data values of *x* and *y* axes, along with the corresponding cell and embryo identifier for each data point when appropriate (cell identifier and embryo name, respectively). Each sheet is labelled with the relevant figure panel number. (XLSB)

**S4 Data. Excel file for graphs in Fig 5.** Data for each graph (A, B, C, C') is given in a separate sheet containing raw data values of *x* and *y* axes, along with the corresponding cell and embryo identifier for each data point when appropriate (cell identifier and embryo name, respectively). Each sheet is labelled with the relevant figure panel number. (XLSB)

**S5 Data. Excel file for graphs in Fig 6.** Data for each graph (A, B, C, D, E, F) is given in a separate sheet containing raw data values of *x* and *y* axes, along with the corresponding cell and embryo identifier for each data point when appropriate (cell identifier and embryo name, respectively). Each sheet is labelled with the relevant figure panel number. (XLSB)

**S6 Data. Excel file for graphs in S3 Fig.** Data for each graph (A, B, C, D, E, F) is given in a separate sheet containing raw data values of *x* and *y* axes, along with the corresponding cell and embryo identifier for each data point when appropriate (cell identifier and embryo name, respectively). Each sheet is labelled with the relevant figure panel number. (XLSB)

**S7 Data. Excel file for graphs in S4 Fig.** Data for each graph (B, C, D, D', E) is given in a separate sheet containing raw data values of *x* and *y* axes, along with the corresponding cell and embryo identifier for each data point when appropriate (cell identifier and embryo name, respectively). Each sheet is labelled with the relevant figure panel number. (XLSB)

**S8 Data. Excel file for graphs in S5 Fig.** Data for each graph (A, A', B, C, D) is given in a separate sheet containing raw data values of *x* and *y* axes, along with the corresponding cell and embryo identifier for each data point when appropriate (cell identifier and embryo name, respectively). Each sheet is labelled with the relevant figure panel number.
(XLSB)

**S9 Data. Excel file for graphs in S6 Fig.** Data for each graph (A, A', C, D) is given in a separate sheet containing raw data values of *x* and *y* axes, along with the corresponding cell and embryo identifier for each data point when appropriate (cell identifier and embryo name, respectively). Each sheet is labelled with the relevant figure panel number.
(XLSB)

**S10 Data. Excel file for graphs in S7 Fig.** Data for graph (E) is given in a separate sheet containing raw data values of *x* and *y* axes, along with the corresponding cell and embryo identifier for each data point when appropriate (cell identifier and embryo name, respectively). The sheet is labelled with the relevant figure panel number.
(XLSB)

## Acknowledgments

We thank Bruno Monier, Stefano De Renzis, and Bloomington Drosophila Stock Center (https://bdsc.indiana.edu/) for *Drosophila* strains. We thank all members of Bénédicte Sanson's research group for discussions.

## Author Contributions

**Conceptualization:** Claire M. Lye, Alexander Nestor-Bergmann, Bénédicte Sanson.

**Formal analysis:** Claire M. Lye.

**Funding acquisition:** Bénédicte Sanson.

**Investigation:** Claire M. Lye, Jenny Evans.

**Methodology:** Claire M. Lye, Guy B. Blanchard.

**Project administration:** Bénédicte Sanson.

**Software:** Guy B. Blanchard.

**Supervision:** Bénédicte Sanson.

**Visualization:** Claire M. Lye, Jenny Evans.

**Writing – original draft:** Claire M. Lye, Bénédicte Sanson.

**Writing – review & editing:** Claire M. Lye, Guy B. Blanchard, Jenny Evans, Alexander Nestor-Bergmann, Bénédicte Sanson.

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
