## [Editor Report · Decision Letter 0]

10 Oct 2023

Dear Dr Sanson, 

Thank you for submitting your manuscript entitled "Drosophila axis extension is robust to an orthogonal pull by invaginating mesoderm" for consideration as a Research Article by PLOS Biology.

Your manuscript, the reviews from Review Commons, and your revision plan have now been evaluated by the PLOS Biology editorial staff as well as by an academic editor with relevant expertise and I am writing to let you know that we are interested in inviting a revision of your study. 

However, before we can formally invite a revision of your manuscript, we need you to complete your submission by providing the metadata that is required for full assessment. To this end, please login to Editorial Manager where you will find the paper in the 'Submissions Needing Revisions' folder on your homepage. Please click 'Revise Submission' from the Action Links and complete all additional questions in the submission questionnaire.

Once your full submission is complete, your paper will undergo a series of checks. After your manuscript has passed the checks, we will send you a 'major revision' decision, wherein the status of your manuscript will be formally switched to 'in revision' and we will pass along any editorial input on the revision plan. 

To provide the metadata for your submission, please Login to Editorial Manager (https://www.editorialmanager.com/pbiology) within two working days, i.e. by Oct 12 2023 11:59PM.

Kind regards,

Luke

Lucas Smith, Ph.D.

Senior Editor

PLOS Biology

lsmith@plos.org

---

## [Editor Report · Decision Letter 1]

16 Oct 2023

Dear Dr Lye,

Thank you again for submitting your manuscript "Drosophila axis extension is robust to an orthogonal pull by invaginating mesoderm" along with reviews from Review Commons, for consideration at PLOS Biology, and thank you for now completing the metadata related to your submission. As mentioned in my last email, your manuscript and revision plan have been assessed by the PLOS Biology editorial team and by an Academic Editor with relevant expertise, and we would like to invite you to revise the work to thoroughly address the reviewers' reports. 

Overall, the Academic Editor has commented that your revision plan is appropriate, and so we encourage you to revise the study as detailed there. Given the extent of revision needed, we cannot make a decision about publication until we have seen the revised manuscript and your response to the reviewers' comments. Your revised manuscript is likely to be sent for further evaluation by all or a subset of the reviewers, and we will ultimately be looking to see that they are satisfied by the changes made.

**IMPORTANT - SUBMITTING YOUR REVISION**

*Re-submission Checklist*

*Published Peer Review*

*PLOS Data Policy*

*Blot and Gel Data Policy*

Sincerely,

Luke

Lucas Smith, Ph.D.

Senior Editor

PLOS Biology

lsmith@plos.org

---

## [Decision Letter · Decision Letter 2]

20 Feb 2024

Dear Dr Sanson,

Thank you for your patience while we considered your revised manuscript "­­­Polarised cell intercalation during Drosophila axis extension is robust to an orthogonal pull by the invaginating mesoderm" for publication as a Research Article at PLOS Biology. This revised version of your manuscript has been evaluated by the PLOS Biology editors, the Academic Editor and the original reviewers.

The reviews are appended below. As you sill see, the reviewers are largely satisfied with the revision. We do note that reviewer 1, while suggesting that we accept the study, has flagged that some points were not fully addressed with new experiments. Having discussed this comment with the Academic Editor, we agree that, while the data requested might be nice to have it is not strictly required for publication. I would also note that Reviewer 3, while highlighting the strengths of the study, wonders whether this would be better suited for PLOS One, given the observational nature of the work. Overall, we do not share this concern, and we think the study is within the scope of our journal. 

Therefore, based on the reviews and our Academic Editor's assessment of the study we are likely to accept this manuscript for publication. However, before we can accept your manuscript, we need you to address the following data and other policy-related requests.

**IMPORTANT: Please address the following editorial requests: 

1) TITLE: We would like to suggest a slight tweak to your title, as we think the use of 'robust' is a bit unclear without reading the paper. We suggest the title be changed to: 

"Polarized cell intercalation during Drosophila axis extension is unaffected by the orthogonal pull generated by the invaginating mesoderm"

2) BLURB: In the relevant section of our online system, please provide a blurb which (if accepted) will be included in our weekly and monthly Electronic Table of Contents, sent out to readers of PLOS Biology, and may be used to promote your article in social media. The blurb should be about 30-40 words long and is subject to editorial changes. It should, without exaggeration, entice people to read your manuscript. It should not be redundant with the title and should not contain acronyms or abbreviations.

3) DATA: You may be aware of the PLOS Data Policy, which requires that all data be made available without restriction: http://journals.plos.org/plosbiology/s/data-availability. For more information, please also see this editorial: http://dx.doi.org/10.1371/journal.pbio.1001797

a. Supplementary files (e.g., excel). Please ensure that all data files are uploaded as 'Supporting Information' and are invariably referred to (in the manuscript, figure legends, and the Description field when uploading your files) using the following format verbatim: S1 Data, S2 Data, etc. Multiple panels of a single or even several figures can be included as multiple sheets in one excel file that is saved using exactly the following convention: S1_Data.xlsx (using an underscore).

b. Deposition in a publicly available repository. Please also provide the accession code or a reviewer link so that we may view your data before publication. 

>>Regardless of the method selected, please ensure that you provide the individual numerical values that underlie the summary data displayed in the following figure panels as they are essential for readers to assess your analysis and to reproduce it:

Fig 2B-K; Fig 3D-H; Fig 4A-B,D; Fig 5A-C; Fig 6A-F; Fig S3A-F; Fig S4B-E; Fig S5A-D; Fig S6A,C; Fig S7E;

>>Please also ensure that figure legends in your manuscript include information on where the underlying data can be found, and ensure your supplemental data file/s has a legend.

>>Please ensure that your Data Statement in the submission system accurately describes where your data can be found.

4) CODE: Per journal policy, if any code was generated to support the conclusions of your manuscript, we would require that you make it available without restrictions upon publication. Please ensure that any code is sufficiently well documented and reusable, and that your Data Statement in the Editorial Manager submission system accurately describes where your code can be found.

We expect to receive your revised manuscript within two weeks. 

*Published Peer Review History*

*Press*

Sincerely,

Lucas

Lucas Smith, Ph.D.

Senior Editor

lsmith@plos.org

PLOS Biology

Reviewer remarks:

Reviewer #1: The authors have provided satisfactory responses to most of my points. Some of the arguments provided to dismiss experimental suggestions are disappointing (e.g. that acquiring a gap43:mCherry and myosin:GFP dataset that could be properly quantified in both channels "is not possible within the timeframe of a revision"). But overall this is a detailed, well-done study.

Reviewer #2: The authors have addressed all my previous questions. I fully support the publication of this elegant piece of work in PLOS Biology.

Reviewer #3: I believe the authors have done an admirable job of responding to all the reviewer comments, including my own, and I think the manuscript is improved because of it. I have no significant comments on the text or figures. As I stated in my earlier review, I admire this paper for its rigor, and I feel that it definitively addresses the question of how forces from the mesoderm affect the ectoderm during Drosophila convergent extension. Performing these analyses in a single twist mutant background with an MRLC knock-in tag is an experimental advance over previously published results, and I think the quantitative methods used to analyze myosin dynamics and cell topology/intercalation are hard to find fault with. Therefore, I believe these results should certainly be published, and will be of interest to groups studying convergent extension in particular, and biomechanics in general. That being said, this work is largely observational, comparing wild-type embryos to just one other mutant background, and nearly all the results reported as significant are marginally significant and/or relatively transient. Therefore, I don't see a problem publishing this manuscript in its current form in PLoS Biology, but it might be more appropriate for a journal such as PLoS ONE.

---

## [Editor Report · Decision Letter 3]

3 Apr 2024

Dear Bénédicte,

Thank you for the submission of your revised Research Article "­­­Polarised cell intercalation during Drosophila axis extension is robust to an orthogonal pull by the invaginating mesoderm" for publication in PLOS Biology, and thank you for addressing our last editorial requests in this revision. On behalf of my colleagues and the Academic Editor, François Schweisguth, I am pleased to say that we can in principle accept your manuscript for publication, provided you address any remaining formatting and reporting issues. These will be detailed in an email you should receive within 2-3 business days from our colleagues in the journal operations team; no action is required from you until then. Please note that we will not be able to formally accept your manuscript and schedule it for publication until you have completed any requested changes.

**As a note - I have updated your data availability statement, in our online system, to reference the DOI you generated related to your code deposition. Please do double check that the new statement is accurate and that everything looks good. 

PRESS

Sincerely, 

Luke

Lucas Smith, Ph.D.

Senior Editor

PLOS Biology

lsmith@plos.org